# The Restorative Potential of Green Cultural Heritage: Exploring Cultural Ecosystem Services' Impact on Stress Reduction and Attention Restoration

**Jing Xie** [1], **Shixian Luo** [2], **Katsunori Furuya** [1,*], **Huixin Wang** [1], **Jiao Zhang** [1], **Qian Wang** [1], **Hongyu Li** [1] and **Jie Chen** [1]

[1]  Graduate School of Horticulture, Chiba University, Chiba 271-8510, Japan;
21hd5402@student.gs.chiba-u.jp (J.X.); wanghuixin1229@yeah.net (H.W.);
20hd5401@student.gs.chiba-u.jp (J.Z.); mayu.wq@gmail.com (Q.W.); aysa5422@chiba-u.jp (H.L.);
cdga0939@chiba-u.jp (J.C.)
[2]  School of Architecture, Southwest Jiaotong University, Chengdu 611756, China; shixianluo@swjtu.edu.cn
*  Correspondence: k.furuya@faculty.chiba-u.jp

**Abstract:** Green cultural heritage is an important form of natural space in cities. Only a few studies have conducted restorative studies in a historical environment as most have focused on natural environments. Moreover, few studies on cultural ecosystem services (CESs) have addressed cultural heritage. Based on an onsite questionnaire distributed to green cultural heritage users (N = 64) in Hamarikyu Garden, this paper explores the value of CESs in a green cultural heritage site and the relationship between cultural ecosystem values and perceived attention restoration/stress reduction. A multiple linear regression analysis and simple linear regression analyses were used to examine the data. The results showed that (1) the cultural ecosystem values of the green cultural heritage site were all rated highly except for the sense of place; (2) spending time in green cultural heritage provided respondents significant perceived attention restoration and stress reduction; (3) aesthetics and cultural heritage significantly affected perceived stress reduction, while attention restoration showed a significant positive correlation with aesthetic value and sense of place; and (4) the more visitors perceived the value of CESs, the more significant the perceived stress reduction and attention recovery were. This study indicates that CESs represent a useful tool for measuring the environmental characteristics of green cultural heritage sites and can predict perceived psychological recovery in green cultural heritage sites. Our findings enhance our knowledge about restorative environmental attributes through objective descriptions of potential health-promoting qualities and can be utilized as inspiration for designing restorative environments in green cultural heritage sites.

**Keywords:** perceived mental recovery; attention restoration theory; stress reduction theory; cultural ecosystem services; green cultural heritage

## 1. Introduction

Cultural heritage refers to the tangible and intangible heritage assets passed down from generation to generation by a group or society [1]. However, not all assets passed through the ages are considered "heritage"; rather, heritage is specifically the product of social choices [2]. When assessing cultural heritage, it is important to consider its intrinsic value in terms of its links with sustainable development [3,4]. This intrinsic value encompasses the ability to convey the values and meanings of the past and serves as a living memory of the local ecosystem [5]. It also conveys the intangible meaning of the heritage site, including the quality of the place and the emotional attachment people develop to it [6].

Scholars in architecture, planning, travel, and other related disciplines have conducted many studies on cultural heritage, such as those related to urban planning in World Heritage Cities [4,7]; the architecture of heritage buildings [8,9]; and heritage tourism [10–12].

However, cultural heritage includes not only buildings and structures but also culturally significant landscapes and natural environments that are often overlooked by scholars.

The inclusion of landscapes in the domain of heritage is the result of an allegorical expansion of heritage throughout the 20th century [13]. Landscape architecture implies an understanding of the concept of territory as a social outcome. It involves the introduction of historical elements that make landscape architecture a product with evolutionary and transformative dimensions [2]. Although landscape architecture is the creator of green cultural heritage (an environment combining nature and history) [14], research on green cultural heritage (GCH) remains limited. The semantic expansion of heritage to include landscapes was introduced [13], giving rise to the concept of green cultural heritage, which emphasizes the importance of preserving natural and cultural landscapes. In addition, green cultural heritage (which is not yet strictly defined), according to Bučas (2006), is a type of heritage created with the use of living natural materials (e.g., plants, flowers) rather than human-made materials; green cultural heritage sites are created by modifying the green elements of nature according to human ideas [14]. This kind of place, where humans and nature coexist harmoniously, is not just about ecological arboriculture; nor is it just about an architectural site. Therefore, these areas should be referred to as green cultural heritage sites, considering their nature and purpose as serious spiritual places conducive to communication and a quiet connection with nature [14]. GCH sites are important green infrastructure in cities, and numerous studies have demonstrated the restorative benefits of green spaces. Several scholarly efforts have been made to study the landscape quality, health, and well-being of CGH sites. For example, Deng et al. (2020) found that the effects of three landscape types and various elements of a traditional park on psychophysiological activities were investigated using physiological and psychological indicators [15]; Luo et al. (2022) investigated restorative experience in the pavilions of a GCH site [16]. However, research on the restorative experience encompassing both the stress reduction and attention restoration effects of GCH sites has been neglected. In addition to the restorative qualities of a green space's natural elements, some studies have begun to explore the perceived restorative benefits that cultural and historical elements provide users; however, very little research has been carried out on green spaces with historical and cultural ambience. Therefore, the effective usage of this neglected green asset in cities can play an important role in the development of healthy cities. Additionally, little attention has been paid to the cultural ecosystem services (CESs) of urban cultural heritage, and established research has overemphasized people's experience of the natural environment in urban spaces rather than the cultural environment. Thus, there is a gap in research on the relationship between restorative experiences and perceived CESs in GCH sites.

Starting from the above considerations, this study aims to examine the restorative potential of GCH sites and explore the impact of CESs on stress reduction and attention restoration; furthermore, the associations between perceived CESs and restorative experiences (perceived stress reduction and perceived attention restoration) were explored. Following the Introduction in Section 1, Section 2 discusses the theoretical background and conceptualization of CESs, the SRT, and the ART and the development of a hypothesis. Section 3 then provides a detailed discussion of the methods used. Section 4 presents the findings of this study, while Section 5 presents a discussion. Finally, Section 6 discusses the conclusion, implications, limitations, and future research suggestions.

## 2. Literature Review

### 2.1. Current Research on Cultural Ecosystem Services Ignores Cultural Heritage

There is consensus that the urban ecosystem services provided by green spaces not only support sustainable urban development but also improve environmental quality and human health and well-being. The Millennium Ecosystem Assessment (MA) was initiated by the United Nations and Global Environment Facility in 2001 to assess the state of the world's ecosystem services and formulate protection programs [17]. To quantify the benefits obtained from ecosystem functions, the MA proposed the concept of ecosystem services

and divided it into four categories: regulating, supporting, supplying, and cultural services. In more detail, nature provides us with water, clean air, and food, as well as raw materials for medicine, industry, and construction. In addition, enjoying parks, landscapes, and wildlife improves our health and well-being. All these benefits are called ecosystem services. Ecosystem services are therefore the direct and indirect contributions that ecosystems make to human well-being and quality of life.

Cultural ecosystem services (CESs), as a subcategory of ecosystem services, focus on the scope of cultural services. CESs are defined as the "non-material" benefits that people derive from ecosystems, such as aesthetic information, recreation, spirituality, enrichment, or cultural heritage [17,18]. Such non-material CESs arise from human interactions with the biophysical environment. Although the importance of CESs is generally recognized, its assessment lags behind that of more specific services [19,20] because its subjective value is intangible. Therefore, the quantification of CESs is difficult in biophysical or monetary terms [21], and its values may vary according to an individual's social and cultural norms [22]. From this perspective, the evaluation of CESs requires extracting complementary capabilities and methods from the social sciences and humanities [23].

Current CES research overemphasizes people's experiences of nature in urban spaces. For example, Kremer et al. (2016) [24] argued that CESs are the most important ecosystem services for urban dwellers because they represent some of people's most familiar and personal experiences of nature in urban environments, which may inspire a willingness to protect the natural environment and influence conservation practices [25]. Furthermore, the CES values provided by urban parks have received widespread attention [17,26,27]. Larson et al. (2016) referred to CES values in urban greenways [28]. Dickinson and Hobbs (2017) focused on the importance of CESs in urban green spaces for urban residents and the role of management in urban green spaces [29]. While some scholars have noted the CESs provided by urban green infrastructure [30], only a few have identified their value potential in cultural heritage and the impact of CES values on human well-being [31] and cultural heritage management [32].

Schaich et al. (2010) argued that the evaluation of CESs is an assessment method that goes beyond purely economic values by integrating social and humanistic approaches, which can establish new research frameworks in the field of heritage [33]. CES assessments can detect features that are beneficial to people from the perspective of cultural landscapes and thus understand their potential for sustainable development [31]. In addition, the management of cultural heritage can gain traction through CES evaluations as they can assess the potential of a site or area in terms of human well-being [18,32]. When considering the current needs of cultural heritage research, it is pertinent to explore the link between cultural heritage and CESs. Despite the emphasis on the benefits of CES research for cultural heritage [2], current research on CESs continues to focus on natural experiences at the expense of cultural landscapes.

### 2.2. Absent Studies of Restorative Potential in GCH

Currently, urban life generates many physical health problems and mental stresses such as disease and cancer. Residents living in large cities (e.g., London, New York, Tokyo) may experience physical and psychological stresses that reduce their personal quality of life [34–36]. Over the past four decades, a growing body of research has shown that urban green spaces are an important resource for public health as they can have a restorative effect on cognitive processes [37] and help relieve stress [38]. In this context, the stress reduction theory (SRT [39]) and attention restoration theory (ART [40]) have emerged as major approaches to explaining the restorative benefits of natural experiences and are frequently used restorative theories today [16,41–43].

For Ulrich et al. (1991), the SRT is a psychological evolutionary model that focuses on the physical environment [39]. It emphasizes the importance of recovery from psychological and physiological stresses associated with threats or challenges based on emotional

functioning [16,44]. Therefore, an individual's experience of relaxation, escape from daily worries, and calmness can be considered a reduction in perceived stress [45].

The ART emphasizes the importance of recovery from attentional fatigue based on cognitive functioning. Kaplan (1995) categorized the restorative characteristics of the ART into four components: fascination, being away, extent, and compatibility, with the understanding that the depletion of directed attention can be restored through exposure to a natural environment [40]. The understanding of the ART distinguishes between two attentional mechanisms of attentional patterns: "directed" and "effortless" [46]. Thus, the state of reduced directed attention fatigue based on cognitive functioning is a restorative experience (subjective-measure outcomes of such experiences can be termed "perceived attention restoration" [44]).

Restoration properties can manifest in a range of environments, from completely natural to fully constructed. Kaplan and Kaplan (1989), for example, clearly stated that a high-level built environment also has some restoration potential [47]. Currently, there are fewer studies on the restoration potential of the built environment than there are on the restoration potential of the natural environment [46]; however, studies examining the built environment are gradually receiving more scholarly attention. Kaplan (1993) found that museum environments have some restorative potential [48]; subsequently, Packer and Bond (2010) found that museums can also have the same restorative qualities as natural environments [49]. Abdulkarim and Nasar (2014) found that certain architectural elements, such as sculptures and seating, also made sites more restorative [41]. Hidalgo et al. (2006) found that people's preferred built environments (historical/cultural and recreational sites) were also restorative [50]. Ulrich et al. (1991) found that natural environments are more restorative than man-made environments in emotional and physical terms [39]. In addition to natural or built environments, the restoration potential of mixed natural and historic environments should also be investigated [51]. Historic green spaces in cities are green cultural heritage sites that blend nature and historic culture. It is worthwhile to investigate the restorative potential for people in environments containing both natural and historical characteristics (i.e., green cultural heritage sites).

### 2.3. Linking the CESs of Green Cultural Heritage Sites to Restorative Experience

Little is known about whether CESs, the measurement of the cultural features of an environment, are associated with restorative experiences in green cultural heritage sites. Several restorative experience studies have examined the correlations between perceived sensory dimensions (PSDs, as characterizations of environmental features) and stress reduction achieved through environmental restoration experiences. Grahn and Stigsdotter (2010) identified eight perceptual sensory dimensions of perception and described the people who reported stress as environmentally sensitive to illustrate the relationship between the perceptual sensory dimensions of an urban green space and perceived stress reduction [42]. Luo et al. (2022) investigated the relationship between PSDs and restorative experience in the pavilions of urban parks [16]. Additionally, Memari et al. (2017) investigated the relationship between PSDs and perceived stress reduction using the Short-Revised Restoration Scale, which is based on theoretical interpretations of the SRT [52]. Peschardt and Stigsdotter (2013) examined the experience of attentional restoration in nine small, public urban green spaces in the city of Copenhagen using the perceived restorativeness scale [53]. Therefore, similar to PSDs, as characterizations of environmental features and properties, this study argues that visitors' perceptions of green cultural heritage CESs are potentially linked to restorative experiences.

### 2.4. Hypothesis Development

Based on the above literature review, the following hypotheses are proposed.

(1)   In a GCH site, individuals can feel CES values.
(2)   Individuals can perceive stress recovery and attention restoration.
(3)   Certain values of CESs can significantly predict recovery potential.

Furthermore, considering the findings of Riechers et al. (2018) on the differences in urban green perception among people with different CES sensitivities [54], we would also like to address the following hypothesis:

(4)　In GCH sites, perceived stress reduction and attention recovery differ for individuals with varying CES sensitivity.

Figure 1 illustrates the framework of this study.

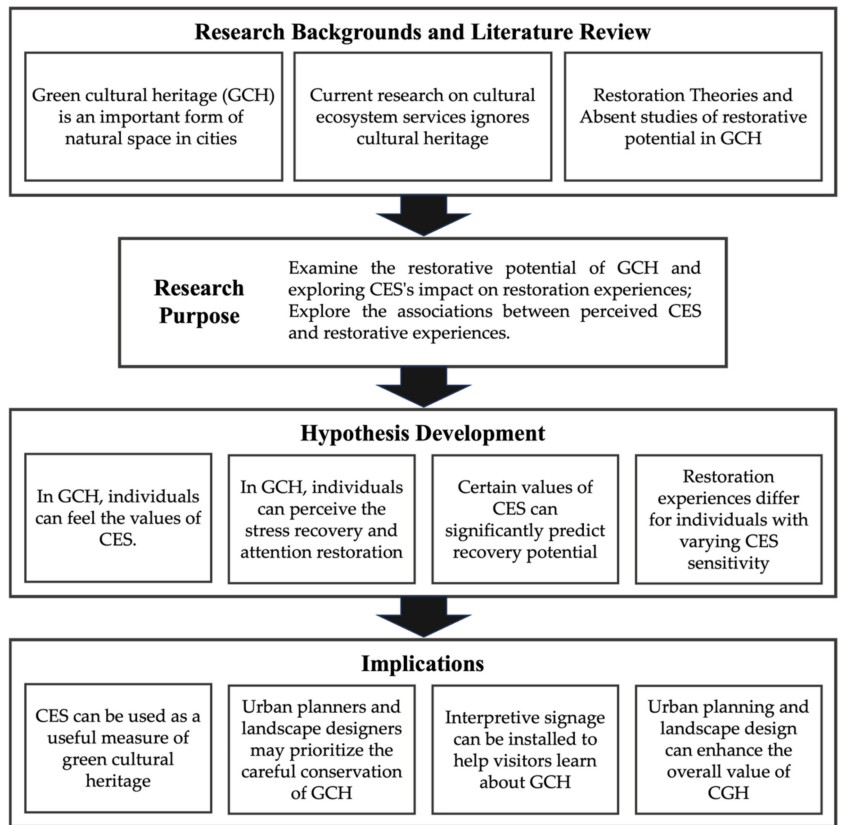

**Figure 1.** The research framework.

### 3. Materials and Methods

*3.1. Study Site*

Hamarikyu Garden (HG) is a metropolitan garden covering an area of 250,165.81 m$^2$. Located in Chuo Ward, at the mouth of the Sumida River along the Tokyo Bay, HG is the only place in Tokyo that has retained its tidal pond. It is one of only nine places in Japan that have been designated as both a Special National Historic Site of Japan and a Special National Place of Scenic Beauty (Figure 2).

During the Edo period, the garden was used as a residence and falconry ground for the Tokugawa shoguns. After the Meiji Restoration, it was used as a detached palace for the imperial family and was later opened as a public garden in April 1946. In Hamarikyu Garden, there are four water bodies: the tidal pond, two Kamoba (duck-hunting sites), and an inner moat. Around the duck-hunting sites, there are konozoki (small openings) through which nobles would observe the ducks while they were eating and being caught as a traditional form of Japanese entertainment. Around the bay, there are four important buildings, the most famous of which is the Najajima-no-ochaya (teahouse), where visitors can rest for a short time. In addition, there are abundant vegetation spaces, such as three-hundred-year-old pine trees, artificially pruned vegetation, and flower seas. Visitors can experience the rich natural environment and cultural atmosphere (Figure 3).

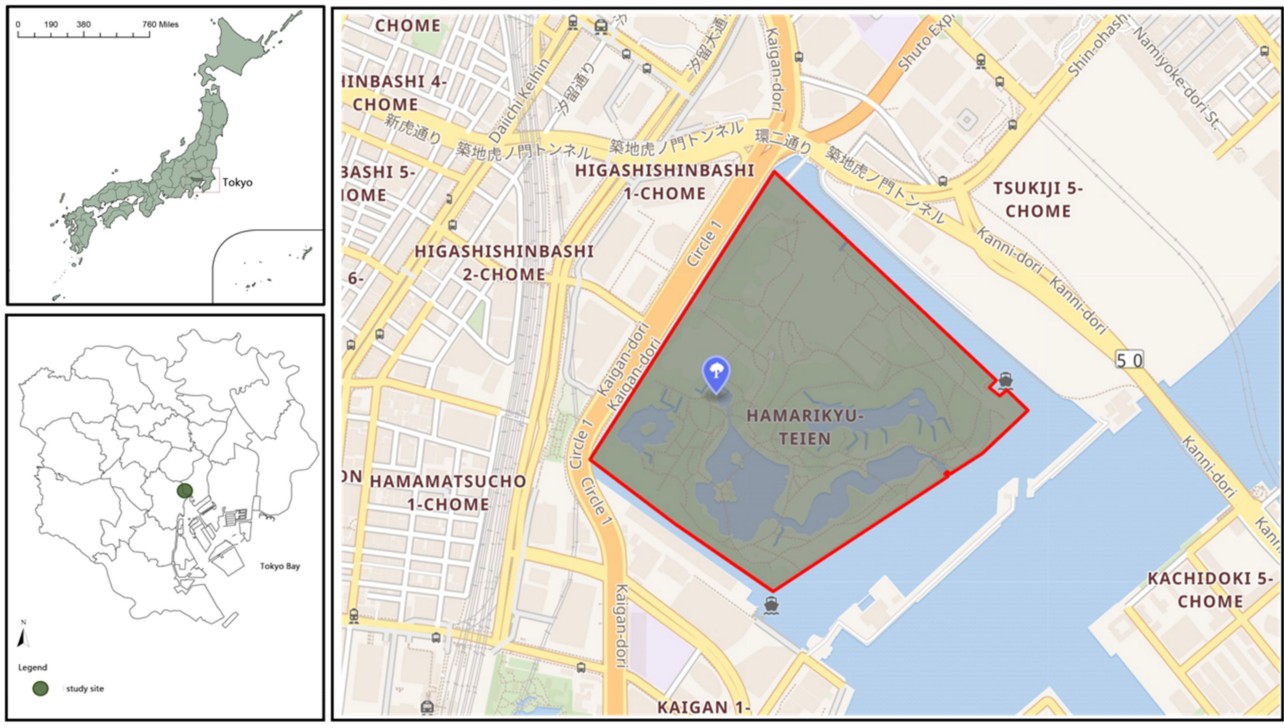

**Figure 2.** Location of the study site (map credit: OpenStreetMap).

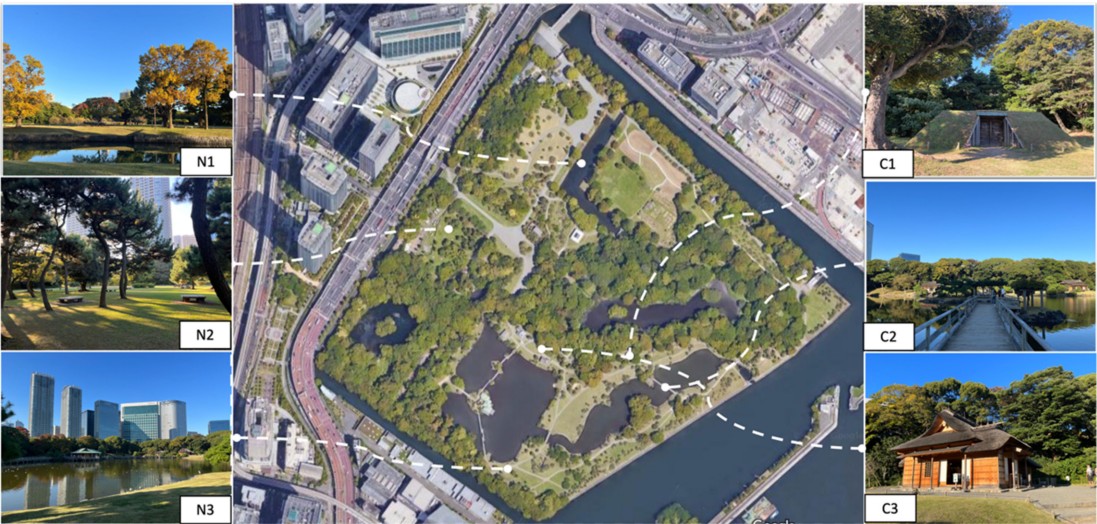

**Figure 3.** Locations of on-site natural images ((**N1**) the inner moat, (**N2**) an open space, and (**N3**) the tidal pond) and cultural images ((**C1**) konozoki, (**C2**) the bridge over the pond, and (**C3**) the teahouse) of HG (map credit: Google Maps).

### 3.2. Participants and Procedures

First, a pilot survey was conducted in September 2022. Six research assistants participated in the pilot survey, after which they became volunteers to guide survey subjects on tours. After the pilot survey and training, these volunteers were very familiar with the experimental sites and processes. The formal on-site survey was conducted in October, and a total of 64 students were recruited as a sample through web announcements and posters at Chiba University. The final version of the survey questionnaire was completed after considering the feedback of the six pilot survey respondents. The final sample included 24 males and 40 females, ranging from 22 to 40 years of age (mean age: 25.8 ± 3.51 years). In terms of their educational background, 26.56% were undergraduate students, 50.00%

were graduate students, and 23.44% were doctoral students. All students were from the Graduate School of Horticulture, majoring in landscape architecture, urban planning, and other majors (environmental science for bioproduction, applied biological chemistry, food and resource economics, etc.).

The formal on-site survey took approximately 2.5 h to complete (Figure 4). It required subjects (groups of one to three people) to first tour the garden for approximately two hours to ensure that they had a sufficient perception of it. Additionally, to ensure that the respondents visited most of the main tour areas of the gardens and experienced similar levels of green cultural heritage environments (cultural, historical, and natural), each group was guided by a research assistant. In addition, during the tour, the research assistants did not make any introductions to the garden to avoid any disruption to the respondents' experiences. Subjects were also forbidden to communicate or interact with each other. After the tour, they completed the questionnaire in a quiet pavilion to prevent their responses from being influenced by external factors. This study was conducted according to the guidelines of the Declaration of Helsinki and approved by the Ethics Committee of Chiba University. However, there was no examination of the human body or physiological data in this study, and all participants were kept anonymous. Therefore, no ethics review was required to be submitted to the ethics committee. All participants signed an informed consent form.

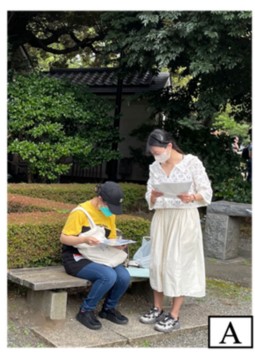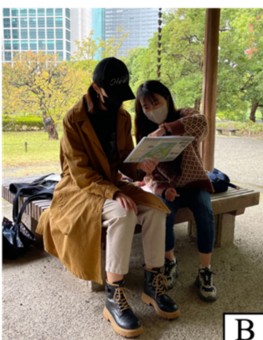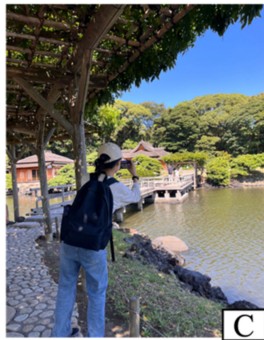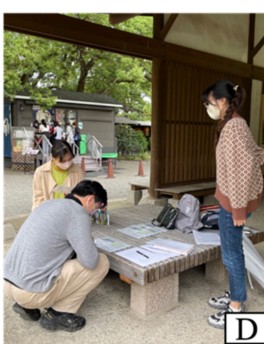

**Figure 4.** The survey procedure. (**A**) Get the brochure of the GCH and go to the pavilion near the entrance; (**B**) Read the brochure and start their tour with a volunteer who would guide the route; (**C**) Take a two-hour tour according to the route; (**D**) Go back to the pavilion to complete the questionnaire after a short break.

### 3.3. Measures

The categories used for the CES values were originally adapted from the types of CESs described in the MA. The number of CES values and the specific indicators constantly change depending on the object and purpose of the study [17,55,56]. To assist with the contextualization and appropriate classification of the CESs, the authors (eight, comprising one professor, one assistant professor, and six Ph.D. students) visited Hamarikyu Garden in June 2022 to gain a comprehensive understanding of the garden's physical surroundings, historical elements, and cultural characteristics. Finally, following Sherrouse et al. (2014) [57], van Riper et al. (2020) [58], and Zhang et al. (2022) [17], after fieldwork and a discussion of the prior literature, the authors screened and identified ten CES dimensions: aesthetics, recreation, social relations, education, nature appreciation, sense of place, bequest, health, therapeutics, and cultural heritage (Table 1). The CESs were all measured using a 7-point Likert scale ranging from 1 (strongly disagree) to 7 (strongly agree).

**Table 1.** Classification categories of CESs.

| CES Values | Description |
|---|---|
| Aesthetics | I think this place has an attractive landscape. |
| Recreation | I think there are opportunities for recreation in this place. |
| Social relations | I can spend time with family, friends, or other people here. |
| Education | I think this place can provide opportunities for scientific research or public education. |
| Nature appreciation | I think this place has a variety of animals, plants, and other living species. |
| Sense of place | I feel a sense of belonging through some certain features here. |
| Bequest | I think this place is likely to be passed down unchanged for generations to come. |
| Health | I think this place makes me feel better physically. |
| Therapeutics | I think this place makes me forget my troubles and helps me feel relaxed. |
| Cultural heritage | I think this place has several historically or culturally significant landscape features. |

Two psychological scales were used to measure the participants' self-reported stress reduction and directed attention restoration outcomes. All outcomes were measured using a 7-point Likert scale ranging from 1 (strongly disagree) to 7 (strongly agree). The participants were first asked about their perceived restoration in terms of "restorative experiences", "positive emotions", and "stress reduction" [45,59]. While being away is one of the components of the ART concept, along with fascination, extent, and compatibility [60], it is more reasonable to use physical components (novelty) and psychological components (escape) instead of the single factor of being away [61]. Thus, the improved perceived attention restoration scale consists of five factors: fascination, novelty, escape, extent, and compatibility. Each factor has three to five items (Table 2).

**Table 2.** Seven-point scale of perceived stress reduction and perceived attention restoration.

| Measurement | Restorative Experiences | Reference |
|---|---|---|
| Stress reduction | I forget daily worries and feel restored here.<br>I feel happy and comfortable here.<br>I feel relaxed and calm here. | [45,59] |
| Novelty | This is an escape from everyday surroundings.<br>I can switch up my daily activities (study/work) here.<br>There are many novelties here. | |
| Escape | Being here makes me feels like an escape from my daily chores.<br>This place takes my mind off the daily tasks I must do for a while.<br>This is a place where I can forget the presence of others and enjoy being alone.<br>I feel liberated from my everyday surroundings here. | |
| Fascination | The environment has a charming quality.<br>My attention was drawn to many interesting things.<br>I want to get to know this place better.<br>There is so much to explore and discover here.<br>I want to spend more time looking around this place. | [60–62] |
| Extent | There are so many troubling things going on here.<br>It is a confusing place to be.<br>There are many distractions here.<br>It is very disorganized here. | |
| Compatibility | I can do what I like here.<br>I feel a sense of belonging here.<br>I feel that I can fit in with this environment.<br>Staying in this place suits my personality.<br>I can find many ways to enjoy the surroundings. | |

### 3.4. Analysis

This study utilized a within-group design. The survey data were compiled and statistically analyzed using Excel software (Version 16.76). All statistical analyses were performed

using JASP 0.16.4, and the level of significance was set at $p < 0.05$. First, the reliability (internal consistency) of the scales was assessed and measured using Cronbach's alpha index; the validity of the two psychological scales were measured using Kaiser–Meyer–Olkin values. Then, simple linear regression analyses were used to examine the linear relationships between restorative experience outcomes and ten CES items. Afterwards, a multiple linear regression analysis (introduced via the method "Enter") was used to explore the CES predictors that significantly affect perceived stress reduction and attention restoration. The independent variables of the above two models were the ten CES factors, and the dependent variables were the perceived stress reduction, overall perceived attention restoration, and five components of the perceived attention restoration. Additionally, based on the degree of the CES values, visitors were divided into two categories in the following analysis: a sensitive group (CES values higher than the mean value) and an insensitive group (CES values lower than or equal to the mean value). An independent samples *t*-test was used to determine the differences in perceived stress reduction and attention restoration between the two groups.

## 4. Results

### 4.1. Assessment of the CES Values and the Perception of Stress Reduction and Attention Restoration

As shown in Figure 5, the 10 CES values of HG were rated highly. All mean values were greater than five points except for sense of place (4.78 ± 1.18). Aesthetics (6.28 ± 1.10), cultural heritage (6.28 ± 0.86), social relations (6.22 ± 0.96), recreation (6.14 ± 0.98), bequest (6.13 ± 1.15), and educational values (6.11 ± 0.97) had mean values greater than six, indicating that they were perceived to the highest degree. These were followed by therapeutics (5.61 ± 1.21), nature appreciation (5.50 ± 1.15), and health (5.47 ± 1.31).

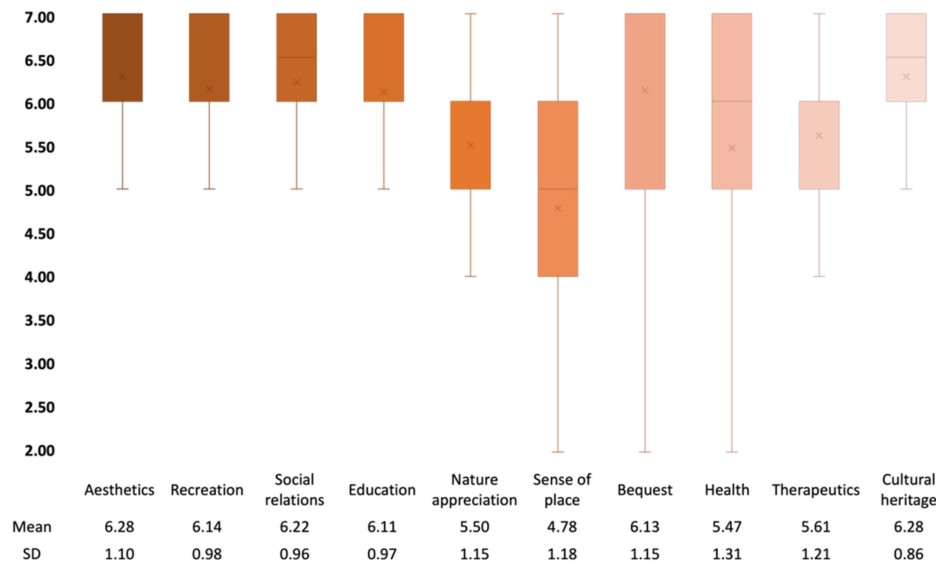

**Figure 5.** Overall assessment of CES values (means ± standard deviations).

For perceived attention restoration, most respondents perceived a high level of attention restoration when they stayed in the green cultural heritage area; the mean score was greater than five except for compatibility (4.64 ± 1.03) (Figure 6). In detail, extent (5.69 ± 0.81) received the highest score, indicating that HG allowed subjects to perceive a high range of exploration and a coherent environment. Extent was followed by fascination (5.43 ± 0.85), which indicates that green cultural heritage can provoke effortless attention from respondents. Escape (5.22 ± 1.15) indicated that the respondents believed that staying in such an environment could make them feel disconnected from their daily work environment. Compatibility (4.64 ± 1.03) received the lowest rating, indicating that the respondents had difficulty matching their internal motivations with the environment.

In addition, the perceived stress reduction (5.83 ± 0.82) was greater than the mean of each component of attention restoration, indicating that the visitors felt a greater reduction in stress in the HG.

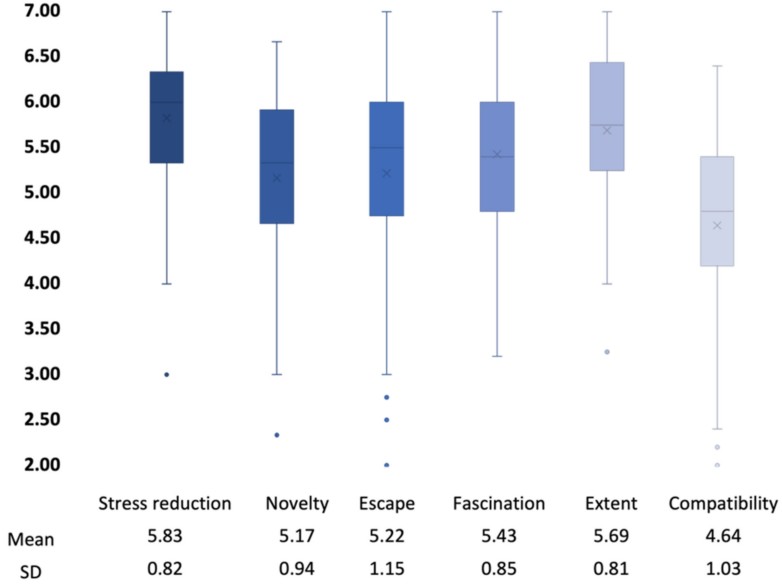

**Figure 6.** Overall assessment of perceived attention restoration components and perceived stress reduction (mean ± standard deviations).

The reliability of the CESs, perceived stress reduction, and attention restoration was verified, and Cronbach's alpha ranged from 0.616 to 0.891; the results showed high internal consistency (Table A1). Moreover, the Kaiser–Meyer–Olkin value for the perceived stress reduction scale was 0.707 ($p < 0.001$) and its value for the perceived attention restoration scale was 0.717 ($p < 0.001$), indicating that the above two psychological scales demonstrated high validity.

*4.2. Association of CES Values with Perceived Stress Reduction and Components of Perceived Attention Restoration*

The results for each CES from our linear regression analysis are shown in Table 3, arranged into one section for perceived stress reduction and another for each component of perceived attention restoration.

**Table 3.** Estimates and statistics of the CESs' effect on perceived stress reduction, novelty, escape, fascination, extent, and compatibility.

| | B | St. Error | Sig. | 95% Confidence Interval | | F | R² |
| --- | --- | --- | --- | --- | --- | --- | --- |
| | | | | Lower | Upper | | |
| *Perceived Stress reduction* | | | | | | | |
| Aesthetics | 0.455 | 0.070 | <0.001 | 0.366 | 0.642 | 53.300 | 0.462 |
| Recreation | 0.490 | 0.085 | <0.001 | 0.321 | 0.659 | 33.460 | 0.351 |
| Social relations | 0.533 | 0.083 | <0.001 | 0.366 | 0.700 | 40.754 | 0.397 |
| Education | 0.164 | 0.133 | 0.222 | −0.102 | 0.431 | 1.519 | 0.024 |
| Nature appreciation | 0.204 | 0.086 | 0.021 | 0.032 | 0.377 | 5.620 | 0.083 |
| Sense of place | 0.284 | 0.080 | <0.001 | 0.124 | 0.443 | 12.669 | 0.170 |
| Bequest | 0.216 | 0.085 | 0.014 | 0.046 | 0.387 | 6.434 | 0.094 |
| Health | 0.117 | 0.077 | 0.136 | −0.038 | 0.251 | 0.271 | 0.035 |
| Therapeutics | 0.262 | 0.079 | 0.002 | 0.104 | 0.419 | 11.037 | 0.151 |
| Cultural heritage | 0.449 | 0.106 | <0.001 | 0.237 | 0.661 | 17.916 | 0.224 |

**Table 3.** *Cont.*

|  | B | St. Error | Sig. | 95% Confidence Interval | | F | R$^2$ |
|---|---|---|---|---|---|---|---|
|  |  |  |  | Lower | Upper |  |  |
| *Novelty* |  |  |  |  |  |  |  |
| Aesthetics | 0.530 | 0.093 | <0.001 | 0.270 | 0.640 | 24.163 | 0.280 |
| Recreation | 0.430 | 0.110 | <0.001 | 0.193 | 0.633 | 14.085 | 0.185 |
| Social relations | 0.245 | 0.108 | <0.001 | 0.269 | 0.703 | 20.114 | 0.245 |
| Education | 0.339 | 0.116 | 0.006 | 0.097 | 0.561 | 8.037 | 0.115 |
| Nature appreciation | 0.198 | 0.102 | 0.117 | −0.042 | 0.368 | 2.533 | 0.039 |
| Sense of place | 0.328 | 0.096 | 0.008 | 0.070 | 0.454 | 7.465 | 0.107 |
| Bequest | 0.235 | 0.101 | 0.062 | −0.010 | 0.394 | 3.616 | 0.055 |
| Health | 0.017 | 0.091 | 0.894 | −0.170 | 0.195 | 0.018 | 0.000 |
| Therapeutics | 0.195 | 0.097 | 0.123 | −0.042 | 0,346 | 2.441 | 0.038 |
| Cultural heritage | 0.316 | 0.133 | 0.011 | 0.083 | 0.613 | 6.900 | 0.100 |
| *Escape* |  |  |  |  |  |  |  |
| Aesthetics | 0.511 | 0.114 | <0.001 | 0.306 | 0.762 | 21.864 | 0.261 |
| Recreation | 0.507 | 0.128 | <0.001 | 0.336 | 0.847 | 21.463 | 0.257 |
| Social relations | 0.500 | 0.131 | <0.001 | 0.334 | 0.860 | 20.657 | 0.250 |
| Education | 0.224 | 0.146 | 0.075 | −0.027 | 0.557 | 3.288 | 0.050 |
| Nature appreciation | 0.271 | 0.122 | 0.030 | 0.026 | 0.515 | 4.904 | 0.073 |
| Sense of place | 0.460 | 0.110 | <0.001 | 0.229 | 0.667 | 16.681 | 0.212 |
| Bequest | 0.207 | 0.124 | 0.101 | −0.041 | 0.453 | 2.776 | 0.043 |
| Health | 0.140 | 0.110 | 0.271 | −0.098 | 0.342 | 1.236 | 0.020 |
| Therapeutics | 0.502 | 0.104 | <0.001 | 0.269 | 0.685 | 20.922 | 0.252 |
| Cultural heritage | 0.188 | 0.167 | 0.137 | −0.082 | 0.585 | 2.273 | 0.035 |
| *Fascination* |  |  |  |  |  |  |  |
| Aesthetics | 0.491 | 0.086 | <0.001 | 0.209 | 0.552 | 19.670 | 0.241 |
| Recreation | 0.530 | 0.093 | <0.001 | 0.273 | 0.646 | 24.239 | 0.281 |
| Social relations | 0.467 | 0.100 | <0.001 | 0.215 | 0.613 | 17.309 | 0.218 |
| Education | 0.375 | 0.103 | 0.002 | 0.122 | 0.535 | 10.143 | 0.141 |
| Nature appreciation | 0.380 | 0.087 | 0.002 | 0.108 | 0.457 | 10.463 | 0.144 |
| Sense of place | 0.271 | 0.088 | 0.030 | 0.019 | 0.372 | 4.915 | 0.073 |
| Bequest | 0.411 | 0.085 | <0.001 | 0.132 | 0.474 | 12.589 | 0.169 |
| Health | 0.002 | 0.082 | 0.986 | −0.163 | 0.166 | 0.000 | 0.000 |
| Therapeutics | 0.335 | 0.084 | 0.004 | 0.083 | 0.417 | 8.920 | 0.126 |
| Cultural heritage | 0.491 | 0.110 | <0.001 | 0.268 | 0.707 | 19.700 | 0.241 |
| *Extent* |  |  |  |  |  |  |  |
| Aesthetics | 0.291 | 0.090 | 0.020 | 0.036 | 0.395 | 5.732 | 0.085 |
| Recreation | 0.256 | 0.101 | 0.042 | 0.008 | 0.414 | 4.332 | 0.065 |
| Social relations | 0.308 | 0.102 | 0.013 | 0.056 | 0.464 | 6.479 | 0.095 |
| Education | 0.127 | 0.105 | 0.316 | −0.104 | 0.317 | 1.021 | 0.016 |
| Nature appreciation | 0.271 | 0.087 | 0.030 | 0.019 | 0.365 | 4.921 | 0.074 |
| Sense of place | 0.363 | 0.081 | 0.003 | 0.087 | 0.412 | 9.380 | 0.131 |
| Bequest | −0.013 | 0.089 | 0.918 | −0.188 | 0.169 | 0.011 | 0.000 |
| Health | 0.338 | 0.074 | 0.006 | 0.061 | 0.357 | 8.008 | 0.114 |
| Therapeutics | 0.160 | 0.084 | 0.206 | −0.061 | 0.276 | 1.633 | 0.026 |
| Cultural heritage | 0.277 | 0.116 | 0.027 | 0.031 | 0.493 | 5.146 | 0.077 |
| *Compatibility* |  |  |  |  |  |  |  |
| Aesthetics | 0.448 | 0.107 | <0.001 | 0.208 | 0.635 | 15.590 | 0.201 |
| Recreation | 0.377 | 0.123 | 0.002 | 0.149 | 0.642 | 10.260 | 0.142 |
| Social relations | 0.316 | 0.129 | 0.011 | 0.080 | 0.598 | 6.869 | 0.100 |
| Education | 0.155 | 0.133 | 0.222 | −0.102 | 0.431 | 1.519 | 0.024 |
| Nature appreciation | 0.392 | 0.105 | 0.001 | 0.142 | 0.562 | 11.241 | 0.153 |
| Sense of place | 0.448 | 0.099 | <0.001 | 0.193 | 0.590 | 15.550 | 0.201 |
| Bequest | 0.016 | 0.114 | 0.897 | −0.212 | 0.242 | 0.017 | 0.000 |
| Health | 0.066 | 0.100 | 0.605 | −0.147 | 0.251 | 0.270 | 0.004 |
| Therapeutics | 0.252 | 0.105 | 0.044 | 0.006 | 0.425 | 4.220 | 0.064 |
| Cultural heritage | 0.245 | 0.148 | 0.051 | −0.002 | 0.590 | 3.944 | 0.060 |

Note: The results show the 10 simple linear regression analyses for evaluating mental restoration, and the bold numbers indicate the CESs significantly associated with perceived stress reduction and components of perceived attention restoration.

For perceived stress reduction, aesthetics (B = 0.455, $p < 0.001$), recreation (B = 0.490, $p < 0.001$), social relations (B = 0.533, $p < 0.001$), cultural heritage (B = 0.449, $p < 0.001$), nature appreciation (B = 0.204, $p = 0.021$), bequest (B = 0.216, $p = 0.014$), and therapeutics (B = 0.262, $p = 0.002$) could significantly help people feel psychologically relaxed and recover from daily stress.

For components of perceived attention restoration, the CESs that significantly impacted novelty were aesthetics (B = 0.455, $p < 0.001$), recreation (B = 0.430, $p < 0.001$), sense of place (B = 0.328, $p = 0.008$), education (B = 0.339, $p = 0.006$), cultural heritage (B = 0.316, $p = 0.011$), and social relations (B = 0.245, $p < 0.001$). This means that HG brought beautiful landscapes, sounds, and smells to the respondents and provided them with a place to spend time outdoors. In addition, the participants' experiences in HG also allowed them to discover and feel a historical and cultural atmosphere.

For escape, the six CES values of aesthetics (B = 0.511, $p < 0.001$), recreation (B = 0.507, $p < 0.001$), social relations (B = 0.500, $p < 0.001$), therapeutics (B = 0.502, $p < 0.001$), nature appreciation (B = 0.271, $p = 0.03$), and sense of place (B = 0.460, $p < 0.001$) contributed to the respondents' significant feelings of being away from their daily environment, including work.

Fascination indicates the attractiveness of an environment, which motivates respondents to explore environmental spaces. According to the results, all nine CES values significantly ($p < 0.05$) influenced the perception of fascination except for health (B = 0.002, $p = 0.986$), which is related to physical exercise. Fascination is related to the attractiveness of the environment, so this result suggests that the higher an individual's perception of the nine CES values, the higher their level of fascination with the GCH as well.

Extent represents the environments with the potential for continued exploration [46,63,64]. The results indicate that sense of place (B = 0.363, $p = 0.003$), social relations (B = 0.308, $p = 0.013$), health (B = 0.338, $p = 0.006$), cultural heritage (B = 0.277, $p = 0.027$), nature appreciation (B = 0.271, $p = 0.030$), aesthetics (B = 0.291, $p = 0.020$), and recreation (B = 0.256, $p = 0.042$) were all significantly associated with extent. Hence, the participants were more likely to explore environments that had a sense of place. Moreover, they were more attracted to spaces where they could engage in social and healthy activities.

Compatibility represents the ability of a place to provide opportunities to engage in activities that are "compatible" with internal motivations. In HG, perceptions of aesthetics (B = 0.448, $p < 0.001$), sense of place (B = 0.448, $p < 0.001$), nature appreciation (B = 0.392, $p = 0.001$), recreation (B = 0.377, $p = 0.002$), social relations (B = 0.316, $p = 0.011$), and therapeutics (B = 0.252, $p = 0.044$) were all significantly correlated with compatibility.

*4.3. CES Items That Significantly Predict Perceived Stress Reduction and Attention Restoration*

Seven multiple linear regressions were conducted to establish the overall relationship between the ten CES values of green cultural heritage and attention recovery and stress reduction. The dependent variables of the seven regression models were perceived stress reduction, overall perceived attention restoration, and five components of attention restoration. The normality of the model residuals (Table 4) was tested using the Kolmogorov–Smirnov test, and the results showed normal distribution ($p > 0.05$).

**Table 4.** Kolmogorov–Smirnov test results.

| | PSR | PAR | Novelty | Escape | Fascination | Extent | Compatibility |
|---|---|---|---|---|---|---|---|
| K-S Z value | 0.146 | 0.134 | 0.132 | 0.167 | 0.065 | 0.884 | 0.104 |
| *p* value | 0.129 | 0.188 | 0.218 | 0.056 | 0.950 | 0.415 | 0.493 |

PSR: perceived stress reduction; PAR: perceived attention restoration.

Arriaza et al. (2004) stated that a model's tolerance value < 0.2 or VIF value > 10 indicates a multicollinearity problem [65]. The current model did not find a multicollinearity problem and was considered acceptable, with the lowest tolerance = 0.476 and the highest VIF = 2.099 (Table 5).

**Table 5.** Significant predictors of perceived stress reduction, perceived attention restoration, and components of perceived attention restoration.

| Dependent Variable | Independent Variable | Unstandardized Beta | Standard Error | Standardized Beta | t | Sig. | Collinearity Statistics | |
|---|---|---|---|---|---|---|---|---|
| | | | | | | | Tolerance | VIF |
| Perceived stress reduction (adjusted R$^2$ = 0.639) | (constant) | 0.422 | 0.663 | | 0.637 | 0.527 | | |
| | Aesthetics | 0.283 | 0.080 | 0.381 | 3.554 | <0.001 | 0.591 | 1.691 |
| | Cultural heritage | 0.183 | 0.091 | 0.192 | 2.002 | 0.050 | 0.737 | 1.357 |
| Perceived attention restoration (adjusted R$^2$ = 0.598) | (constant) | 0.969 | 0.537 | | 1.803 | 0.077 | | |
| | Aesthetics | 0.219 | 0.064 | 0.353 | 3.399 | 0.001 | 0.591 | 1.691 |
| | Sense of place | 0.173 | 0.055 | 0.300 | 3.120 | 0.003 | 0.692 | 1.445 |
| Novelty (adjusted R$^2$ = 0.303) | (constant) | 0.839 | 0.980 | | 0.856 | 0.396 | | |
| | Aesthetics | 0.281 | 0.118 | 0.327 | 2.388 | 0.021 | 0.591 | 1.691 |
| Escape (adjusted R$^2$ = 0.403) | (constant) | 0.077 | 1.103 | | 0.070 | 0.944 | | |
| | Aesthetics | 0.286 | 0.132 | 0.274 | 2.165 | 0.035 | 0.591 | 1.691 |
| | Sense of place | 0.262 | 0.115 | 0.270 | 2.306 | 0.025 | 0.692 | 1.445 |
| | Therapeutics | 0.295 | 0.134 | 0.311 | 2.203 | 0.032 | 0.476 | 2.099 |
| Fascination (adjusted R$^2$ = 0.420) | (constant) | 0.206 | 0.807 | | 0.256 | 0.799 | | |
| | Cultural heritage | 0.274 | 0.111 | 0.276 | 2.469 | 0.017 | 0.737 | 1.357 |
| Compatibility (adjusted R$^2$ = 0.292) | (constant) | 1.018 | 1.080 | | 0.943 | 0.350 | | |
| | Aesthetics | 0.309 | 0.130 | 0.329 | 2.387 | 0.021 | 0.591 | 1.691 |
| | Sense of place | 0.256 | 0.111 | 0.292 | 2.295 | 0.026 | 0.692 | 1.445 |

Note: No CES item was a significant predictor of extent, so this component was omitted.

As shown in Table 5, in terms of stress recovery, the values of aesthetic (B = 0.283, *p* < 0.001) and cultural heritage (B = 0.183, *p* = 0.05) were able to significantly influence stress recovery, with these two factors explaining 63.9% of the variance. In terms of attention recovery, aesthetics (B = 0.219, *p* = 0.001) and sense of place (B = 0.173, *p* = 0.003) positively influenced attention restoration, with the two explaining 59.8% of the variance. This result suggests that the higher the level of perceived environmental aesthetics among green cultural heritage sites, the greater the perceived restoration of attention and stress relief. Furthermore, attention to the historical atmosphere of the site was effective at reducing stress, and a higher sense of connection to the site significantly increased participants' levels of attention.

### 4.4. Differences in Perceived Stress Reduction and Attention Restoration among People with Different CES Sensitivities

By calculating the mean value of the ten CESs assessed by all respondents ($M_{CES}$ = 5.85), the participants were divided into a sensitive group (sensitive to CES perception (mean > 5.85)) and an insensitive group (not sensitive to CES perception (mean ≤ 5.85)). The Mann–Whitney U-test was performed to explore whether there was a significant difference between the perceived mental restoration of the two groups (Figure 7). Attention recovery was significantly lower in the insensitive group (4.763 ± 0.672) than in the sensitive group (5.571 ± 0.462) (*p* < 0.001). In addition, perceived stress reduction was lower in the insensitive group (5.271 ± 0.832) than in the sensitive group (6.234 ± 0.521) (*p* < 0.001).

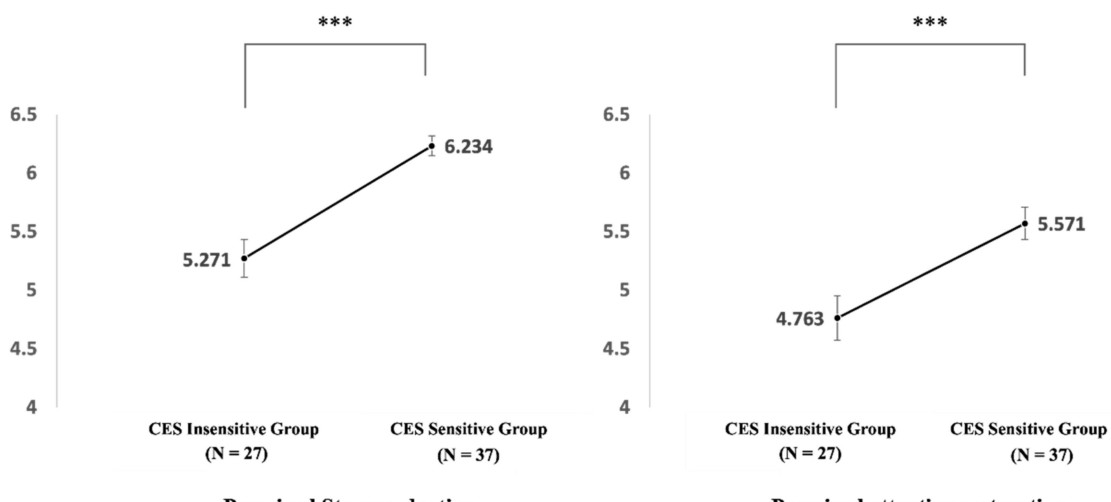

**Figure 7.** Differences in restorative experiences between the sensitive and insensitive groups (*** *p* < 0.001).

## 5. Discussion

Based on an onsite questionnaire distributed to green cultural heritage users in Hamarikyu Garden, this paper explores the values of CESs in a green cultural heritage site and the relationship between cultural ecosystem values and perceived attention restoration/stress recovery. The main results are as follows: (1) all nine cultural ecosystem values of green cultural heritage except for sense of place received high ratings, especially the values aesthetic, cultural heritage, social relations, recreation, bequest, and education; (2) most of the CES values are associated with perceived stress reduction and the components of perceived attention restoration, respectively; (3) the values of aesthetics and cultural heritage significantly affected stress restoration, while attention restoration showed a significant positive correlation with the aesthetic and sense of place values; and (4) the more visitors perceived the CES values, the more significant their perceived stress reduction and attention recovery were. Both expected and unexpected results were found. The possible explanations are discussed below.

### 5.1. Representation of CES Values in Green Cultural Heritage

Compared to other green spaces such as parks, greenways, and forests in the city, green cultural heritage sites have more intangible historical, cultural, educational, and even legacy value. For example, there are many very Japanese landscape expressions, such as traditional tea house buildings, shrines, konozoki (small openings), a bridge over the pond, the pond, open space, water gates, pine trees, and red leaves at the study site (Figure 3). These landscape elements create a scene that combines Japanese history and culture, allowing people to feel high levels of some CES values such as the historical and cultural dimensions. Surprisingly, however, the overall rating for sense of place was low, perhaps due to the fact that the chosen site is a setting of an environment dominated by natural elements and historical elements that are very different from the daily urban environment in which the students live (modern and human-made). According to the results of Section 3.1, our findings support the argument that green cultural heritage provides multiple interactive benefits. In general, all CES values were considered quite important, especially in terms of aesthetics, social relations, recreation, bequest, cultural heritage, and educational values.

It should be noticed that aesthetic and social values are always considered to be very important in green spaces. Specifically, in a survey of a large sample of Berlin city dwellers, aesthetic and social relation values were considered the most important CESs in urban green spaces, which is similar to our findings [54].

Recreation is one of the numerous benefits that individuals and societies gain from landscapes [66]. Compared to other findings, the experiential value provided by greenways has been emphasized in studies of urban greenways, highlighting the importance of recreational connections [28]; similarly, Beckmann-Wübbelt et al. (2021) reported that recreational values were the most desired in forests [67], which is consistent with our findings.

However, bequest, cultural heritage, and educational values are unique compared to other studies which focused on other types of green spaces. For example, in a study by Zhang et al. (2022), bequest did not score very high, even in urban parks with historical and cultural features [17]. Cultural heritage values were considered the least important [28,54]. In addition, cultural heritage and educational values were scored the lowest in forests according to Beckmann-Wübbelt et al. (2021) [67]. These results from previous studies contradict our findings. Green cultural heritage itself is a combination of natural and cultural heritage as it represents the association of humans and nature as an equal community and is considered an organically formed cultural landscapes [14]. Bučas (2006) argued that purposefully formed cultural landscapes give meaning to the results of a person's prioritized activities and show the supremacy of humanity's creative activity over nature [14].

Hence, we can see that the CES values represented in green cultural heritage are not only the regular values which scored very high in other green spaces, such as aesthetics, social relations, recreation, but also some unique values such as bequest, cultural heritage, and educational value.

### 5.2. Driving Factors for the Restorative Experience of Green Cultural Heritage

Restoration theories are usually related to perceived environmental safety [68], attractiveness [43], sensory dimensions, and preferences [53,69]. However, this study considered the drivers of recovery from the perspective of perceived cultural ecosystem services. Our results (Table 5) suggest that aesthetic and cultural heritage values are significant predictors of stress reduction; aesthetics and sense of place significantly influence attention restoration.

Aesthetic appreciation is an important factor for people to feel restored in green spaces [60,70,71]. In line with the results of previous studies [16,72,73], the present study suggests the importance of aesthetic value as an indicator of stress and attention recovery. Consistent with Kaplan (1995) [40], the aesthetic quality of a landscape can act as a soft charm that contributes to recovery from directed attention fatigue; attention restoration, escape, compatibility, and novelty were positively correlated with aesthetic value (Table 5). Therefore, we suggest that methods used to improve aesthetic value can be used to improve the restoration quality.

Cultural heritage values are predictors of stress reduction. This result is consistent with previous research findings [49,59] that historical and cultural values have the potential to offer visitors a restorative experience that provides respite from the stress of daily life. Additionally, as shown in Table 5, they are also important indicators of feelings of fascination. Masullo et al. (2021) found that the dimension most influenced by the component aspects of history and culture is fascination [74]. Therefore, preserving and enhancing the value of cultural heritage is a key proposition in the future of green cultural heritage.

Sense of place is also a reliable predictor of attention recovery potential. Ulrich (1983) asserted the role of memory in cognitive and affective assessments of natural environments and hypothesized a relationship between recovery potential and positive past experiences crystallized in memory [75]. Such memories of positively valanced content have been shown to alleviate negative emotions [76]. Ratcliffe and Korpela (2016) argued that the extent to which people consider a place to be part of themselves and more desirable than any other place for their purpose can be an emotional perception of cognition and indicate potential for recovery and restoration [77].

Thus, the value of being able to bring fond memories or have self-identity within a sense of place can be valuable in the design and management of the physical environments of restorative spaces.

*5.3. More Perception of CESs Leads to a Better Restorative Experience*

Similar to the findings of Riechers et al. (2018) [54], who found differences in the experience results of urban green among people with different CES sensitivities, our results found that people with different sensitivities in perceiving the CES values also produced significant differences in restorative experiences (Figure 7). In other words, people's levels of perception of CESs influence the extent of their self-reported mental recovery (e.g., perceived stress reduction, perceived attention restoration). Previous restorative-related studies have often emphasized that certain landscape elements [15] or their proportions [78] lead to restoration, ignoring the environmental characteristics formed by the elements as a whole. CESs refer to the non-material benefits that people obtain from ecosystems, and they have direct influences on the quality of an individual's life [20]. Our study examined ten CES dimensions in GCH: aesthetics, recreation, social relations, education, nature appreciation, sense of place, bequest, health, therapeutics, and cultural heritage. Thus, the average value of the ten CES dimensions, as an individual's overall perception of the non-material benefits (natural, cultural, historical, educational, etc.) received from a given ecosystem, was used to explore the differences in restorative experiences between populations with different levels of CES perception. As can be seen from Table 4 (regression analysis results), we found that the CES values of aesthetics, cultural heritage, sense of place, and therapeutics all have a significant impact on restorative experience. The mechanisms through which these indicators drive mental restoration were discussed in the previous subsection. Moreover, other studies found that entertainment [79], nature appreciation [46], education [80], and health value [81] will affect people's restorative experience in an environment to a certain extent. Nawrath et al. (2022) found that cultural background affects individuals' experience of green spaces as a source of attention restoration and stress reduction effects [82]. Additionally, studies have been conducted in which people perceive different strengths of attraction with different perceived species richness. Perceiving richer species diversity was found to facilitate recovery from stress [83]. Therefore, in summary, these findings may explain why the higher an individual's overall perceived CES value, the higher their quality of self-reported mental restoration is. This differential perception of mental recovery may be an important means of answering the question of how to construct more restorative environments.

## 6. Conclusions

Although a number of studies have pointed to the benefits of natural environments or historical elements for the quality of people's perceived restoration, green cultural heritage has been neglected for a long time. As a natural public space in the city, the value of a green cultural heritage site lies in its offering of a long-term historical view, questioning its role in people's memory and identity. However, a green cultural heritage site also has an important heritage value that lies precisely in its nature as a humanized and living space, that is, providing a place for relaxation and recovery from mental fatigue.

Few CES studies have addressed cultural heritage. The results of this study suggest that green cultural heritage has the attributes of cultural heritage, education, and bequest. Aesthetic and cultural heritage significantly affect perceived stress reduction, while attention recovery shows a significant positive relationship with aesthetic value and sense of place. The more visitors perceive the value of CESs, the more significant the perceived stress reduction and attention recovery will be. Thus, the value of CESs can be related to perceived psychological recovery in green cultural heritage. Through an objective description of potential health-promoting qualities, our findings can be utilized as inspiration for designing natural restorative environments. Furthermore, our results demonstrate the reliability and validity of using CESs for evaluating the environmental characteristics of green cultural heritage sites. Their application in future cultural heritage research should be encouraged.

The research implications of this study are as follows. First, CESs can be used as useful measures of green cultural heritage. Scholars did not to adequately mention the value

of culture and history among perceptual factors in previous assessments of perceptions of green space. Nolin (2019) argued that in the Anthropocene era, any natural place has cultural value because it has a human imprint, especially in urban environments [84]. Regarding the cultural dimension of PSDs, Peschardt and Stigsdotter (2013) described it as an environment with cultural characteristics such as fountains, sculptures, or exotic plants [53], while Luo et al. (2022) described it as having many human-made elements [16]. The experience of human-fashioned, human-made components, such as fountains, statues, and canals, are referred to as culturally relevant elements [85] and inherent physical relics. In contrast to the simple cultural dimension, cultural heritage is a tangible and intangible asset passed down through the generations [1]. It emphasizes not only the inherent temporal dimension of all physical remains but also the role of heritage as a source of historical knowledge, which people in different periods have used as a tool to learn more about the ways of life of previous generations [86]. The three values of cultural heritage in CESs, bequest, and education, coincide with the characteristics of cultural heritage. In the previous evaluation system, it was difficult to observe the presentation and evaluation of these values.

Second, the findings highlight the importance of preserving GCH sites, not only for their historical and cultural significance but also for their potential to provide mental and emotional benefits to the community. Urban planners and landscape designers may prioritize the careful conservation of such sites to maintain their unique qualities that contribute to stress reduction and attention restoration. Additionally, in recognition of these sites' potential as stress-reducing and attention-restoring spaces, they could be developed as cultural and recreational hubs within cities. This could involve creating pedestrian-friendly areas, providing seating, and enhancing access to the site, encouraging people to spend more time there and engage with its cultural value.

Third, to help visitors and locals understand a green cultural heritage site's significance and its potential impact on mental well-being, interpretive signage can be installed. This signage could explain the historical context, emphasize the importance of the site for mental restoration, and encourage mindful exploration. Additionally, the successful integration of the stress-reducing and attention-restoring elements with a site's cultural heritage requires collaboration between urban planners and heritage experts. This collaboration can lead to innovative designs that preserve the site's authenticity while enhancing its positive impact on mental well-being.

Last, by recognizing the unique qualities of a GCH site as a stress-reducing and attention-restoring space, urban planning and landscape design can enhance the overall value and importance of such sites in a city. The careful consideration of historical preservation and cultural significance, alongside the well-being of residents and visitors, creates an opportunity to create culturally rich and mentally nourishing urban environments.

Several limitations apply to this study. The current scope of green cultural heritage is still vague, and our study chose only one site and just 64 respondents, which may lead to some limitations, and a new round of evaluation is necessary in the future to increase the findings' validity. Second, there were some limitations in recruiting college students. However, the decision to recruit college students as participants was based on specific considerations for experimental cost. The recruitment of an adult, non-student population requires significant funding in Japan; we did not have sufficient funding, and therefore recruited only a student sample in this exploratory experiment. Additionally, Japan was still in the midst of a pandemic at the time of the survey, so it became very difficult to openly recruit samples, whereas it was relatively easy to organize college students from the university to conduct the survey. Therefore, data from different groups could provide various insights for this topic, and future studies that include participants of different ages, nationalities, occupations, residence times, and cultural backgrounds may be more valuable. Third, additional psychological state scales (e.g., the profile of mood states and anxiety) as well as physiological indicators (e.g., blood pressure, blood oxygen levels, and skin conductance levels) could be applied to measure psychological recovery due to

visiting green cultural heritage sites. Additionally, due to study design and conditioning limitations, we only collected data on subjective perceptions of restorative experiences, which may have some potential bias, although this was a common methodology used in previous studies. Future studies could, for example, consider measuring objective attentional improvement through pre-visit versus post-visit attention test scores. Moreover, a comparison of differences in the restoration experiences between green cultural heritage sites and general green spaces would be valuable. Furthermore, due to time and cost limitations, VR usage could be an effective approach to gathering data in a short time period in future studies.

**Author Contributions:** Conceptualization, J.X.; methodology, J.X. and S.L.; software, J.X.; validation, J.X. and S.L.; formal analysis, J.X.; investigation, J.X., S.L., H.W., J.Z., Q.W., H.L. and J.C.; resources, J.X. and S.L.; data curation, J.X. and S.L.; writing—original draft preparation, J.X.; writing—review and editing, J.X., S.L., H.W., J.Z., Q.W., H.L. and J.C.; visualization, J.X.; supervision, K.F.; project administration, K.F.; funding acquisition, J.X. and K.F. All authors have read and agreed to the published version of the manuscript.

**Funding:** This work was supported by the Japan Science and Technology Agency (JST SPRING, grant number JPMJSP2109).

**Data Availability Statement:** The original data used in this study are available upon request.

**Acknowledgments:** We would like to thank all experimental participants for their precious time in this study. Additionally, we thank Miss Nakamura, Mr. Qifeng Cai, and Mr. Sihan Zhang for their help in data collection for this study. Moreover, we thank the reviewers and editors for their valuable time in reviewing this manuscript.

**Conflicts of Interest:** The funders had no role in the design of the study; in the collection, analyses, or interpretation of data; in the writing of the manuscript; or in the decision to publish the results.

## Appendix A

**Table A1.** The internal consistency results.

| Items | Cronbach's $\alpha$ | 95% CI Lower Bound | 95% CI Upper Bound |
|---|---|---|---|
| CESs | 0.788 | 0.696 | 0.856 |
| Stress recovery | 0.790 | 0.682 | 0.865 |
| Attention restoration | 0.875 | 0.824 | 0.914 |
| Novelty | 0.701 | 0.545 | 0.809 |
| Escape | 0.853 | 0.783 | 0.904 |
| Fascination | 0.891 | 0.736 | 0.880 |
| Extent | 0.616 | 0.425 | 0.752 |
| Compatibility | 0.840 | 0.767 | 0.893 |

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
