# Peer review of "The Restorative Potential of Green Cultural Heritage: Exploring Cultural Ecosystem Services’ Impact on Stress Reduction and Attention Restoration"

_forests, doi:10.3390/f14112191_

Round 1

Reviewer 1 Report

Comments and Suggestions for Authors

According to on-site questionnaire surveys of green cultural heritage users, this research investigates the significance of CES within green cultural heritage and the connection between cultural ecosystem values and perceived attention restoration/stress reduction. The findings enhance our understandings of restorative environmental characteristics with objective descriptions of potential health-promoting features, providing inspiration for the design of restorative environments in green cultural heritage.

This research presented in the manuscript is both significant and fascinating. However, there are some manuscript concerns that need further enhancement.

1. It is recommended to clearly differentiate the three sections in terms of content and structure, with "introduction," "literature review," and "motivation and hypothesis development" before the "method" section.

2. It is essential to include a concise introduction to the concepts, dimensions, and relevant studies of the measured variables in the literature review section.

3. The introductory section would benefit from a paragraph explaining how the paper is organized.

4. The title at Line 200 should be “method”, not “introduction”. Please check the entire manuscript and correct similar errors.

5. The sample size of 64 participants for the questionnaire survey is relatively small, and it is recommended to provide more detailed background and demographic information about the participants.

6. It is recommended to provide details regarding the experimental design in method, including independent variables, dependent variables, control variables, between-group/within-group designs, etc.

7. In the "Measures" section, please briefly introduce the existing reliability and validity of the two psychological scales.

8. It is advisable to provide details regarding the CES, including the scoring methodology used to measure the variables.

9. What is the rationale for the standards of 5.40 (line 295) and 4.60 (line 304)?

10. It is recommended to employ Structural Equation Modeling (SEM) to analyze the relationships among numerous independent and dependent variables, concurrently accounting for mediating and moderating variables.

Comments on the Quality of English Language

Overall, the manuscript is readable, but it requires some formatting adjustments and proofreading to enhance its English language quality. I recommend conducting a thorough proofreading to address issues such as grammar errors.

Author Response

Response to Reviewer 1 Comments

We would like to thank you for careful and thorough reading of this manuscript and for the thoughtful comments and constructive suggestions, which help to improve the quality of this manuscript. Here is a point-by-point response to your comments and concerns. All page and line numbers refer to the revised manuscript file.

OVERALL IMPRESSION                            

According to on-site questionnaire surveys of green cultural heritage users, this research investigates the significance of CES within green cultural heritage and the connection between cultural ecosystem values and perceived attention restoration/stress reduction. The findings enhance our understandings of restorative environmental characteristics with objective descriptions of potential health-promoting features, providing inspiration for the design of restorative environments in green cultural heritage.

This research presented in the manuscript is both significant and fascinating. However, there are some manuscript concerns that need further enhancement.

Point 1:

It is recommended to clearly differentiate the three sections in terms of content and structure, with "introduction," "literature review," and "motivation and hypothesis development" before the "method" section.

 Response 1:

Thank you for pointing this out.

Based on your suggestions, the manuscript has been revised into six sections. Following the Introduction in Section 1, the Section 2 discusses the theoretical background and conceptualization of CES, SRT, ART, and the hypothesis development. The Section 3 then provides a detailed discussion of the methods used. The Section 4 presents the findings, while the Section 5 presents the discussion. Finally, after integrating this comment and those of other reviewers, the Section 6 discusses the conclusion, implications, limitations, and future research suggestions.

Please refer to the revised manuscript for details.

Point 2:

It is essential to include a concise introduction to the concepts, dimensions, and relevant studies of the measured variables in the literature review section.

 Response 2:

Thank you for pointing this out.

Based on your suggestions, the Section 2 (Literature Review) mainly introduces the concept of CES and related research, as well as some common CES values, as well as the emergence and development of restorative theories and related research. Based on these literature contents, four hypotheses of this study were proposed.

Please refer the Section 2 "Literature Review" for details.

Point 3:

The introductory section would benefit from a paragraph explaining how the paper is organized.

 Response 3:

Thank you for this helpful comment. The manuscript has been supplemented with a paragraph explaining how the manuscript is organized.

Details are as follows:

(Line 92-96) Following the Introduction in Section 1, the Section 2 discusses the theoretical background and conceptualization of CES, SRT, ART, and the hypothesis development. The Section 3 then provides a detailed discussion of the methods used. The Section 4 presents the findings, while the Section 5 presents the discussion. Finally, the Section 6 discusses the conclusion, implications, limitations, and future research suggestions.

Point 4:

The title at Line 200 should be “method”, not “introduction”. Please check the entire manuscript and correct similar errors.

 Response 4:

Thank you for carefully checking. We are very sorry for this mistake during writing the manuscript, and we have modified it and checked the whole manuscript to ensure no such error again.

Point 5:

The sample size of 64 participants for the questionnaire survey is relatively small, and it is recommended to provide more detailed background and demographic information about the participants.

 Response 5:

Thank you for pointing this out.

Agree with this thoughtful comment. Truly, there are some accurate limitations of recruiting 64 college students. However, we would like to provide some explanations.

First, the decision to recruit college students as participants was based on specific considerations in experimental cost. Because recruiting an adult population other than students in Japan requires significant funding to support, we did not have sufficient funding, and therefore we recruited only a student sample in this exploratory experiment.

Second, Japan was still in the midst of a pandemic at the time we conducted that survey, so it became very difficult to openly recruit samples, whereas it was relatively easy to organize college students from the university to conduct the survey.

Third, despite potential bias, prior experiments have suggested that college students are generally active users of urban space and often have comprehensive knowledge and diverse perspectives. Thus, the college student sample is considered a valid proxy for the general public population, and their experiences and perceptions can provide valuable insights into the topic of green cultural heritage.

Here are some references:

-Jiang, Y., & Yuan, T. (2017). Public perceptions and preferences for wildflower meadows in Beijing, China. Urban Forestry & Urban Greening, 27, 324-331.

-Luo, S., Shi, J., Lu, T., & Furuya, K. (2022). Sit down and rest: Use of virtual reality to evaluate preferences and mental restoration in urban park pavilions. Landscape and Urban Planning, 220, 104336.

-Deng, L., Li, X., Luo, H., Fu, E. K., Ma, J., Sun, L. X., ... & Jia, Y. (2020). Empirical study of landscape types, landscape elements and landscape components of the urban park promoting physiological and psychological restoration. Urban Forestry & Urban Greening, 48, 126488.

-Van den Berg, A. E., Koole, S. L., & van der Wulp, N. Y. (2003). Environmental preference and restoration:(How) are they related?. Journal of environmental psychology, 23(2), 135-146.

-Wilkie, S., & Clements, H. (2018). Further exploration of environment preference and environment type congruence on restoration and perceived restoration potential. Landscape and Urban Planning, 170, 314-319.

-Wang, H., Luo, S., & Furuya, K. (2022). What attracts tourists to press the shutter in cultural heritage tourism? An analysis of visitor-employed photography and visual attributes: a case study on Japan’s Kairakuen Garden. Tourism Recreation Research, 1-17.

However, some accurate limitations of recruiting 64 college students, but we believe that our findings could still offer valuable insights and contribute to the existing knowledge. Also, in the limitations section of the manuscript, we note out and encourage future studies to include a more diverse set of participants to expand the generalizability of the findings.

Furthermore, according to your suggestion, their demographic information (age, gender, and education background) has been additionally included at Subsection 3.2.

Details are as follows:

(Line 258-264) The final sample included 24 males and 40 females, ranging from 22 to 40 years of age (mean age: 25.8 ± 3.51 years). In terms of their educational background, 26.56% are undergraduate student, 50.00% graduate student, and 23.44% are doctor student. All students are from Graduate school of horticultural, majoring in landscape architecture, urban planning, and other major (environmental science for bioproduction, applied biological chemistry, and food and resource economics etc.).

Point 6:

It is recommended to provide details regarding the experimental design in “method”, including independent variables, dependent variables, control variables, between-group/within-group designs, etc.

 Response 6:

Thanks for pointing this out.

Information on our independent and dependent variables, as well as the experimental design (between/within groups) is added in detail in subsection 3.4.

Details are given below:

(Line 308-319) This study utilized a within-group design. The survey data were compiled and statistically analyzed using Excel software. All statistical analyses were performed using JASP 0.16.4, and the level of significance was set at p < 0.05. First, the reliability (internal consistency) of the scales was assessed, as measured by the Cronbach's alpha index; the validity of the two psychological scales were measured by Kaiser–Meyer–Olkin value. Then, the simple linear regression analyses were used to examine the linear relationships between restorative experience outcomes and ten CES items. Afterwards, the multiple linear regression analysis (introduced method as “Enter”) was used to explore the CES predictors that significantly affect perceived stress reduction and attention restoration. The independent variables of the above two models are the ten CES factors, and the dependent variables are perceived stress reduction, overall perceived attention restoration, and five components of the perceived attention restoration.

In addition, because of the within-group design we used, we did our best to consider and control for potential factors that could affect the dependent variable, and we simply assumed that these subjects were a similar sample of individuals. This approach has been implemented in many studies, such as:

- Wang, R., & Zhao, J. (2019). A good sound in the right place: Exploring the effects of auditory-visual combinations on aesthetic preference. Urban Forestry & Urban Greening, 43, 126356.

- Hao, L., Li, D., Songlin, J., Erkang, F., Jun, M., Lingxia, S., ... & Xi, L. (2021). Elements and element components of the rural landscape in Linpan of Western Sichuan in relation to perception, preference and stress recovery. Journal of Resources and Ecology, 12(3), 384-396.

- Luo, S., Xie, J., Wang, H., Wang, Q., Chen, J., Yang, Z., & Furuya, K. (2023). Natural Dose of Blue Restoration: A Field Experiment on Mental Restoration of Urban Blue Spaces. Land, 12(10), 1834.

Therefore, the factors (e.g., age, major) were not included in the statistical modeling.

Point 7:

In the "Measures" section, please briefly introduce the existing reliability and validity of the two psychological scales.

 Response 7:

Thanks for this comment. In order to make the results section coherent, we calculated the reliability and validity of the questionnaire and show these results at the end of subsection 4.1.

The details are given below:

(Line 351-355) The reliability of CES, perceived stress reduction and attention restoration was verified, and Cronbach's alpha ranged from 0.616 to 0.891; the results showed high internal consistency (Table A1). Moreover, the Kaiser–Meyer–Olkin value of perceived stress reduction scale was 0.707 (p < 0.001), perceived attention restoration scale was 0.717 (p < 0.001), indicating that the above two psychological scales were high validity.

Point 8:

It is advisable to provide details regarding the CES, including the scoring methodology used to measure the variables.

 Response 8:

Thanks for pointing this out. Our revised manuscript shown the scoring methodology used to measure the CES values.

Details are as follows:

(Line 289-294) Finally, following Sherrouse et al. (2014) [57], van Riper et al. (2020) [58], and Zhang et al. (2022) [17] did, after fieldwork and discussion of prior literature, the authors screened and identified ten CES dimensions: aesthetics, recreation, social relations, education, nature appreciation, sense of place, bequest, health, therapeutics, and cultural heritage (Table 1). The CES were all measured using a 7-point Likert scale ranging from 1 (strongly disagree) to 7 (strongly agree).

Point 9:

What is the rationale for the standards of 5.40 (line 295) and 4.60 (line 304)?

 Response 9:

Thank you for pointing this out.

As stated in this comment these two values are suddenly, and we apologize for the unclear statement. Therefore, in order to make these two thresholds more convincing, we changed them uniformly to a score of 5 because the scale is a 7-point scale, where a 4 is neutral while a 5 represents a tendency to assess positively.

Details are as follows:

(Line 327-328) As shown in Figure. 5, the 10 CES values of HG were rated highly. All mean values were greater than 5 points except for sense of place (4.78±1.18).

(Line 335-337) For perceived attention restoration, most respondents perceived a high level of attention restoration when they stayed in the green cultural heritage area; the mean score was greater than 5 except for compatibility (4.64±1.03) (Figure. 6).

Point 10:

It is recommended to employ Structural Equation Modeling (SEM) to analyze the relationships among numerous independent and dependent variables, concurrently accounting for mediating and moderating variables.

 Response 10:

Thank you for this thoughtful comment.

The structural equation modeling is a very useful measurement method. Since the main purpose of this study was to find the relationship between CES and perceived recovery, as well as the differences in the degree of restoration among people with different CES sensitivities, we did not further use other analytical methods given the space constraints. However, thank you for your useful recommendations, and given that this was an exploratory study, additional methods of analysis will be taken into account in future large-scale investigations.

Point 11:

Comments on the Quality of English Language

Overall, the manuscript is readable, but it requires some formatting adjustments and proofreading to enhance its English language quality. I recommend conducting a thorough proofreading to address issues such as grammar errors.

 Response 11:

Thank you for reviewing and evaluating the language of our manuscript, we have used a language editing service to revise our manuscript. We hope that the revised manuscript will meet your requirements and expectations.

Additional clarifications

In addition to the above comments, all spelling and grammatical errors have been corrected. Besides, we checked the language of the manuscript again to ensure it was written correctly.

Once again, we thank you for the time you put in reviewing our paper and look forward to meeting your expectations. Since your inputs have been precious, in the eventuality of a publication, we would like to acknowledge your contribution explicitly.

Reviewer 2 Report

Comments and Suggestions for Authors

The study has the potential to contribute to the field, but it requires some improvements in terms of clarity, organization, and explicit connection between theoretical foundations and research questions. It is important to include specific results and implications in the abstract for a more engaging read.

Title

The title is informative but somewhat long. You might consider making it more concise and focused, while still conveying the core message of your study.

For example, you could consider something such as 1- "Restorative Potential of Green Cultural Heritage: Linking Cultural Ecosystem Services and Psychological Well-being."

2- "Restorative Potential of Green Cultural Heritage: Exploring Cultural Ecosystem Services' Impact on Stress Reduction and Attention Restoration."

Abstract:

-        The abstract is well-structured and gives a clear overview of what the study aims to achieve. However, it can be more concise. Try to summarize each point briefly.

-        Line 18 all nine cultural ecosystem values in green cultural heritage? You didn’t mention that you used 9 culture ecosystems. I missed information about the materials and methods in the abstract section.

-        Consider stating the number of participants to provide context for the results.

-        The abstract gives a good overview but lacks specific results. Mention key findings briefly, without going into detail.

-        Mention the key implications of the findings, which can engage readers better.

-        Consider rephrasing the following sentence for clarity: "These are neither ecological arboricultural objects where man and nature coexist in harmony, nor architectural sites." The meaning is not immediately clear.

Introduction:

-        The introduction provides a good background and context for the study.

-        Major: it's quite lengthy, and you could consider shortening it while maintaining the key points.

-        Consider rephrasing this sentence for clarity: "While the natural environment has a higher restoration potential than the built environment with historical elements, the restoration potential of mixed natural and historic environments should also be investigated." This sentence could be more concise and clearer.

-        Clarify the primary research question early in the introduction to guide the reader better.

-        Make sure the introduction aligns clearly with the study's objectives and hypotheses.

-        Line 61. Current research on cultural ecosystem services ignores cultural heritage, please mention how your study addresses this research gap. What makes it unique?

-        Absent studies of restorative potential in GCH: This section provides crucial context for the study, but it could be presented more concisely.

-        Mention how your research adds to the existing body of knowledge, especially in the context of GCH.

Materials and Methods

-        Line 200 2. Introduction “I think it should be Materials and Methods."

-        Clarify the timeline of the study. You mention a pilot survey in September 2022 and a formal survey in October, but it's unclear how much time elapsed between these stages and the subsequent analysis.

Results:

-        The results section is comprehensive.

-        Provide a brief interpretation of the results within the Results section to set the stage for the Discussion. This will help readers understand the significance of your findings without having to wait until the Discussion section.

Discussion:

-        Please clarify the reasons why certain CES values received higher ratings. For example, discuss specific elements of the garden or historical aspects that may have contributed to the high ratings.

-        Address any unexpected results and attempt to explain them. For example, why was the sense of place rated lower than other CES values? Could site characteristics or the sample have contributed to this?

-        Elaborate on the practical implications of your findings. How can urban planners, heritage preservation authorities, or landscape designers use this information to enhance the quality of restorative environments in urban areas?

-        Address the potential limitations of your study. You mentioned using a student sample; discuss how this may impact the generalizability of your results and any other sources of bias. Transparency about limitations is important in research.

Author Response

Response to Reviewer 2 Comments

We would like to thank you for careful and thorough reading of this manuscript and for the thoughtful comments and constructive suggestions, which help to improve the quality of this manuscript. Here is a point-by-point response to your comments and concerns. All page and line numbers refer to the revised manuscript file.

OVERALL IMPRESSION

The study has the potential to contribute to the field, but it requires some improvements in terms of clarity, organization, and explicit connection between theoretical foundations and research questions. It is important to include specific results and implications in the abstract for a more engaging read.

Part 1: Title

Point 1:

The title is informative but somewhat long. You might consider making it more concise and focused, while still conveying the core message of your study.

For example, you could consider something such as:

1- "Restorative Potential of Green Cultural Heritage: Linking Cultural Ecosystem Services and Psychological Well-being."

2- "Restorative Potential of Green Cultural Heritage: Exploring Cultural Ecosystem Services' Impact on Stress Reduction and Attention Restoration."

Response 1:

Thank you for this useful comment about the title. We are highly agreeing that the title should be concise and focused. Therefore, we have changed the title to "Restorative Potential of Green Cultural Heritage: Exploring Cultural Ecosystem Services' Impact on Stress Reduction and Attention Restoration."

Part 2: Abstract

The abstract is well-structured and gives a clear overview of what the study aims to achieve. However, it can be more concise. Try to summarize each point briefly.

 Point 2:

Line 18 all nine cultural ecosystem values in green cultural heritage? You didn’t mention that you used 9 culture ecosystems. I missed information about the materials and methods in the abstract section.

Response 2:

Thanks for point this out. We apologize for this unclear statement, and we rewritten this statement in the revised version.

Details are as follows:

The results showed that 1) the cultural ecosystem values in the green cultural heritage were all rated highly except for the sense of place...

In addition, information on methodology is added to the abstract.

Details are given below:

Based on an onsite questionnaire distributed to green cultural heritage users (N=64) in Hamarikyu Garden, this paper explores the value of CES in green cultural heritage and the relationship between cultural ecosystem values and perceived attention restoration/stress reduction. The multiple linear regression analysis and the simple linear regression analyses were used to examine the data.

Point 3:

Consider stating the number of participants to provide context for the results.

Response3:

Thank you for pointing this out, and the information on sample size is included in the revised abstract.

Details are as follows:

Based on an onsite questionnaire distributed to green cultural heritage users (N=64) in Hamarikyu Garden, this paper explores the value of CES in green cultural heritage and the relationship between cultural ecosystem values and perceived attention restoration/stress reduction.

Point 4:

The abstract gives a good overview but lacks specific results.

Response 4:

Thank you for this comment. Due to the word limit of the abstract, it is hard for us to replicate all the results into the abstract, so only the main findings are given in the current version of the abstract and they are numbered.

Point 5:

Mention key findings briefly, without going into detail. Mention the key implications of the findings, which can engage readers better.

Response 5:

Thank you for this helpful comment. The current version of the abstract shows only the main findings and the implications of the study in question are written at the end.

Part 3: Introduction

The introduction provides a good background and context for the study.

Point 6:

Consider rephrasing the following sentence for clarity: "These are neither ecological arboricultural objects where man and nature coexist in harmony, nor architectural sites." The meaning is not immediately clear.

Response 6:

Thank you for pointing this out. The current sentence is indeed misleading. According to the English Language Correction, we have changed this sentence to: (Line 65-66) “This kind of place, where man and nature coexist harmoniously, is not just about ecological arboriculture, nor is it just about architectural sites.

Point 7: Major

It's quite lengthy, and you could consider shortening it while maintaining the key points.

Response 7:

Agree with this comment. Based on your comments and those of other reviewers, we have made major adjustments to the manuscript. The revised manuscript adds literature review section, and have made some additions and shortenings to retain the key points that need to be stated in the study.

Point 8:

Consider rephrasing this sentence for clarity: "While the natural environment has a higher restoration potential than the built environment with historical elements, the restoration potential of mixed natural and historic environments should also be investigated." This sentence could be more concise and clearer.

Response 8:

Thank you for pointing this out. The current sentence is not easy for reading and I changed it into: (Line189-190) “In addition to natural or built environments, the restoration potential of mixed natural and historic environments should also be investigated.

Point 9:

Clarify the primary research question early in the introduction to guide the reader better.

Response 9:

Agree with this comment. The revised manuscript clarifies the purpose of the study in the penultimate paragraph of the Introduction section.

Details are as follows:

(Line 88-91) Starting from the above considerations, this study aims to examine the restorative potential of GCH and exploring CES's impact on stress reduction and attention restoration; furthermore, the associations between perceived CES and restorative experiences (perceived stress reduction and perceived attention restoration) were explored.  

Point 10:

Make sure the introduction aligns clearly with the study's objectives and hypotheses.

Response 10:

Thank you for pointing this out. Taking into account all the reviewers' comments, we split the background chapter into an introduction chapter and a literature review chapter, and ended the literature review chapter with the research hypotheses of this study.

The details are as follows:

(Line 213-221) Based on the above literature review, the following hypotheses are proposed.

1) In GCH, individuals can feel the values of CES.

2) individuals can perceive the stress recovery and attention restoration.

3) Certain values of CES can significantly predict recovery potential.

Furthermore, considering the findings of Riechers et al. (2018) on the differences in urban green perception among people with different CES sensitivities [61], we would also like to address the following hypothesis:

4) Perceived stress reduction and attention recovery differ for individuals with varying CES sensitivity in GCH.

Point 11:

Line 61. Current research on cultural ecosystem services ignores cultural heritage, please mention how your study addresses this research gap. What makes it unique?

Response 11:

Thank you for this insightful comment.

CES can be used as a useful measure of green cultural heritage. Scholars have yet to adequately mention the value of culture and history among perceptual factors in previous assessments of green space perception.

In addition, in the third paragraph of the conclusion chapter we make a detailed statement:

(Line 653-667) Nolin (2019) argued that in the Anthropocene era, any natural place has cultural value because it has a human imprint, especially in urban environments [85]. In terms of the cultural dimension of PSD, Peschardt and Stigsdotter (2013) described it as an environment with cultural characteristics such as fountains, sculptures, or exotic plants [53], while Luo et al. (2022) described it as having many man-made elements [16]. The experience of human-shaped man-made components, such as fountains, statues, and canals, are referred to as culturally relevant elements [86] and inherent physical relics. In contrast to the simple cultural dimension, cultural heritage is a tangible and intangible asset passed down through the generations [1]. It emphasizes not only the inherent temporal dimension of all physical remains, but also the role of heritage as a source of historical knowledge, which people of different periods have used as a tool to learn more about the way of life of previous generations [87]. The three values of cultural heritage, bequest, and education in CES coincide with the characteristics of cultural heritage. In the previous evaluation system, it was difficult to observe the presentation and evaluation of these values.

Point 12:

Absent studies of restorative potential in GCH. This section provides crucial context for the study, but it could be presented more concisely.

Response 12:

Many thanks for this comment. However, as some reviewers strongly suggested that we add a literature review chapter to detail the context of the study, this subsection has been rewritten and moved to subsection 2.2. Please refer the current version of the subsection 2.2 for details.

Point 13:

Mention how your research adds to the existing body of knowledge, especially in the context of GCH.

Response 13:

The conclusion of the article includes the implication part, which mainly explains how my research adds to the existing body of knowledge, especially in the context of GCH. First, this study demonstrates that CES can be used as a measure of the GCH environment, but not only the green space. Second, the finding highlights the importance of preserving GCH, not only for their historical and cultural significance but also for their potential to provide mental and emotional benefits to the community. Third, we know how to enhance the urban well-being through green cultural heritage.

We mention these values and meanings at the end of the manuscript. Details are as follows:

(Line 650-689) First, CES can be used as a useful measure of green cultural heritage. Scholars have yet to adequately mention the value of culture and history among perceptual factors in previous assessments of green space perception. Nolin (2019) argued that in the Anthropocene era, any natural place has cultural value because it has a human imprint, especially in urban environments [93]. In terms of the cultural dimension of PSD, Peschardt and Stigsdotter (2013) described it as an environment with cultural characteristics such as fountains, sculptures, or exotic plants [60], while Luo et al. (2022) described it as having many man-made elements [41]. The experience of human-shaped man-made components, such as fountains, statues, and canals, are referred to as culturally relevant elements [94] and inherent physical relics. In contrast to the simple cultural dimension, cultural heritage is a tangible and intangible asset passed down through the generations [1]. It emphasizes not only the inherent temporal dimension of all physical remains, but also the role of heritage as a source of historical knowledge, which people of different periods have used as a tool to learn more about the way of life of previous generations [96]. The three values of cultural heritage, bequest, and education in CES coincide with the characteristics of cultural heritage. In the previous evaluation system, it was difficult to observe the presentation and evaluation of these values.

Second, the finding highlights the importance of preserving GCH, not only for their historical and cultural significance but also for their potential to provide mental and emotional benefits to the community. Urban planners and landscape designers may prioritize the careful conservation of such sites to maintain their unique qualities that contribute to stress reduction and attention restoration. Besides, recognizing these sites’ potential as stress-reducing and attention-restoring spaces, they could be developed as cultural and recreational hubs within cities. This could involve creating pedestrian-friendly areas, providing seating, and enhancing access to the site, encouraging people to spend more time there and engage with its cultural value.

Third, to help visitors and locals understand the green cultural heritage site's significance and its potential impact on mental well-being, interpretive signage can be installed. This signage could explain the historical context, emphasize the importance of the site for mental restoration, and encourage mindful exploration. Besides, successful integration of the stress-reducing and attention-restoring elements with the site's cultural heritage requires collaboration between urban planners and heritage experts. This collaboration can lead to innovative designs that preserve the site's authenticity while enhancing its positive impact on mental well-being.

Last, by recognizing the unique qualities of a GCH as a stress-reducing and attention-restoring space, urban planning and landscape design can enhance the overall value and importance of such sites in a city. The careful consideration of historical preservation and cultural significance, alongside the well-being of residents and visitors, creates an opportunity to create culturally rich and mentally nourishing urban environments.

Part 4: Materials and Methods

Point 14:

Line 200 2. Introduction “I think it should be Materials and Methods."

Response 14:

Thanks for the careful examination of our manuscript, and this error has been revised.

Point 15:

Clarify the timeline of the study. You mention a pilot survey in September 2022 and a formal survey in October, but it's unclear how much time elapsed between these stages and the subsequent analysis.

Response 15:

Thank you for this comment. The pilot experiment was conducted in early September, and we then spent one month to recruit volunteers. After the survey was done, the processing and analysis of the data was done in November.

Part 5: Results

Point 16: The results section is comprehensive. Provide a brief interpretation of the results within the Results section to set the stage for the Discussion. This will help readers understand the significance of your findings without having to wait until the Discussion section.

Response 16:

We are very grateful for this professional comment.

We attempted to provide some brief explanations and discussions of these findings in the Results section in the initial version, however, we found that this organization largely reduced the fluency of the reading of the Results section. Therefore, we would like to keep the main discussion and explanations in the Discussion section to maintain the readability of the Results section.

Part 6: Discussion

Point 17:

Please clarify the reasons why certain CES values received higher ratings. For example, discuss specific elements of the garden or historical aspects that may have contributed to the high ratings.

Response 17:

Thanks for this comment. In response to this comment, we make some brief additions to this in subsection 5.1.

Details are as follows:

(Line 458-465) Compared to other green spaces such as parks, greenways and forests in the city, green cultural heritage has more intangible historical, cultural, educational, and even legacy value. For example, there are many very Japanese landscape expressions, such as traditional tea house buildings, shrines, konozoki (small openings), bridge over the pond, the pond, open space, water gates, pine trees, and red leaves in the study site (Figure 3). These landscape elements create a scene combines the Japanese history and culture that allows people to feel some high CES values level, such as the dimensions of historical and cultural.

Point 18:

Address any unexpected results and attempt to explain them. For example, why was the sense of place rated lower than other CES values? Could site characteristics or the sample have contributed to this?

Response 18:

Thanks for this insightful comment. We have added some relevant discussion in subsection 5.1.

Details are given below:

(Line 465-468) Surprisingly, however, the overall rating for sense of place was low, perhaps due to the fact that the chosen site is a setting for an environment dominated by natural elements and historical elements that are very different from the daily urban environment in which the students live (modern, man-made).

Point 19:

Elaborate on the practical implications of your findings. How can urban planners, heritage preservation authorities, or landscape designers use this information to enhance the quality of restorative environments in urban areas?

Response 19:

Agree with this comment. Implication should be a very important part of the research. So, we organized a certain part about the implication in Conclusion section.

Details are as follows:

(Line 650-689) There are the following main research implications in this study. First, CES can be used as a useful measure of green cultural heritage. Scholars have yet to adequately mention the value of culture and history among perceptual factors in previous assessments of green space perception. Nolin (2019) argued that in the Anthropocene era, any natural place has cultural value because it has a human imprint, especially in urban environments [93]. In terms of the cultural dimension of PSD, Peschardt and Stigsdotter (2013) described it as an environment with cultural characteristics such as fountains, sculptures, or exotic plants [60], while Luo et al. (2022) described it as having many man-made elements [41]. The experience of human-shaped man-made components, such as fountains, statues, and canals, are referred to as culturally relevant elements [94] and inherent physical relics. In contrast to the simple cultural dimension, cultural heritage is a tangible and intangible asset passed down through the generations [1]. It emphasizes not only the inherent temporal dimension of all physical remains, but also the role of heritage as a source of historical knowledge, which people of different periods have used as a tool to learn more about the way of life of previous generations [96]. The three values of cultural heritage, bequest, and education in CES coincide with the characteristics of cultural heritage. In the previous evaluation system, it was difficult to observe the presentation and evaluation of these values.

Second, the finding highlights the importance of preserving GCH, not only for their historical and cultural significance but also for their potential to provide mental and emotional benefits to the community. Urban planners and landscape designers may prioritize the careful conservation of such sites to maintain their unique qualities that contribute to stress reduction and attention restoration. Besides, recognizing these sites’ potential as stress-reducing and attention-restoring spaces, they could be developed as cultural and recreational hubs within cities. This could involve creating pedestrian-friendly areas, providing seating, and enhancing access to the site, encouraging people to spend more time there and engage with its cultural value.

Third, to help visitors and locals understand the green cultural heritage site's significance and its potential impact on mental well-being, interpretive signage can be installed. This signage could explain the historical context, emphasize the importance of the site for mental restoration, and encourage mindful exploration. Besides, successful integration of the stress-reducing and attention-restoring elements with the site's cultural heritage requires collaboration between urban planners and heritage experts. This collaboration can lead to innovative designs that preserve the site's authenticity while enhancing its positive impact on mental well-being.

Last, by recognizing the unique qualities of a GCH as a stress-reducing and attention-restoring space, urban planning and landscape design can enhance the overall value and importance of such sites in a city. The careful consideration of historical preservation and cultural significance, alongside the well-being of residents and visitors, creates an opportunity to create culturally rich and mentally nourishing urban environments.

Point 20:

Address the potential limitations of your study. You mentioned using a student sample; discuss how this may impact the generalizability of your results and any other sources of bias. Transparency about limitations is important in research.

Response 20:

Agree with this comment. The student sample alone is indeed one of the limitations of this study. However, due to various limitations (funding, time, COVID-19) this study is currently exploratory. Therefore, we emphasize this potential limitation in the Limitations section of the manuscript and suggest future directions for expansion, such as increasing sample size and sample diversity.

Details are as follows:

(Line 690-702) Several limitations apply to this study. The current scope of green cultural heritage is still vague, and our study chose only one site and just 64 respondents which may lead to some limitations, and a new round of evaluation is necessary in the future to increase validity. Second, there are some limitations of recruiting college students. However, the decision to recruit college students as participants was based on specific considerations in experimental cost. The recruitment of an adult non-student population requires significant funding in Japan; we did not have sufficient funding, and therefore recruited only a student sample in this exploratory experiment. Besides, Japan was still in the midst of a pandemic at the time of the survey, so it became very difficult to openly recruit samples, whereas it was relatively easy to organize college students from the university to conduct the survey. Therefore, data from different groups could provide various insights for this topic, and future studies that include participants of different ages, nationalities, occupations, residence times, and cultural backgrounds may be more valuable.

Additional clarifications

In addition to the above comments, all spelling and grammatical errors have been corrected. Besides, we checked the language of the manuscript again to ensure it was written correctly.

Once again, we thank you for the time you put in reviewing our paper and look forward to meeting your expectations. Since your inputs have been precious, in the eventuality of a publication, we would like to acknowledge your contribution explicitly.

Reviewer 3 Report

Comments and Suggestions for Authors

This paper explores the value of cultural ecosystem services in green cultural heritage and the relationship between cultural ecosystem values and perceived attention restoration/stress reduction. Knowledge restorative environmental attributes through objective descriptions of potential health-promoting qualities and cultural ecosystem services are a useful tool to measure the environmental characteristics of the green cultural heritage, which is the subject of research.
I do recommend the following.

1. The research framework should be significantly adjusted.
Section 2 is “Materials and Methods”, not “Introduction”, and “Literature Review” should be in this part. The final part is “Conclusions”, the managerial implications, the limits of the study, the future research could be in this section.

2. Include a diagram where you explain graphically the methodology used to obtain the information, from theoretical perspective to methodological strategies. On the other hand, the article does not clearly state the purpose of the research.

3. A claim of the relations of variables shall be supported by empirical evidence. The article lacks the hypotheses and supports the conceptual model.

4. Review your references and confirm that you have at least 60% related to articles or other types of document that have no more than 5-6 years of publication. If there is not this proportion, please include newer references.

5. In your conclusions, green cultural heritage has the attributes of cultural heritage, education, and bequest. Aesthetic and cultural heritage significantly affect perceived stress reduction, while attention recovery shows a significant positive relationship with aesthetic value and sense of place. The results demonstrate the reliability and validity of cultural ecosystem services as a method of evaluating the environmental characteristics of green cultural heritage sites. Also, include future research and how you intend to increase the impact of your work.

Comments on the Quality of English Language

Moderate editing of English language required.

Author Response

Response to Reviewer 3 Comments

We would like to thank you for careful and thorough reading of this manuscript and for the thoughtful comments and constructive suggestions, which help to improve the quality of this manuscript. Here is a point-by-point response to your comments and concerns. All page and line numbers refer to the revised manuscript file.

OVERALL IMPRESSION

This paper explores the value of cultural ecosystem services in green cultural heritage and the relationship between cultural ecosystem values and perceived attention restoration/stress reduction. Knowledge restorative environmental attributes through objective descriptions of potential health-promoting qualities and cultural ecosystem services are a useful tool to measure the environmental characteristics of the green cultural heritage, which is the subject of research.
I do recommend the following.

Point 1:

The research framework should be significantly adjusted.
Section 2 is “Materials and Methods”, not “Introduction”, and “Literature Review” should be in this part. The final part is “Conclusions”, the managerial implications, the limits of the study, the future research could be in this section.

Response 1:

Thank you for pointing this out. We are very sorry for this mistake, and we have modified it and checked the whole manuscript to ensure it won't be again. Besides, the overall structure of the manuscript has also been adjusted and currently divided into six sections (Introduction, Literature Review, Materials and Methods, Results, Discussion, and Conclusions) according to this comment.

Point 2:

Include a diagram where you explain graphically the methodology used to obtain the information, from theoretical perspective to methodological strategies. On the other hand, the article does not clearly state the purpose of the research.

Response 2:

Agree with this comment. In the revised version we have inserted a figure at the end of Chapter 2 to illustrate the framework of this study. Besides, the revised manuscript clarifies the purpose of the study in the penultimate paragraph of the Introduction section.

Details are as follows:

(Line 88-91) Starting from the above considerations, this study aims to examine the restorative potential of GCH and exploring CES's impact on stress reduction and attention restoration; furthermore, the associations between perceived CES and restorative experiences (perceived stress reduction and perceived attention restoration) were explored.

Point 3:

A claim of the relations of variables shall be supported by empirical evidence. The article lacks the hypotheses and supports the conceptual model.

Response 3:

Thank you for this helpful comment. The study object is GCH, and the subsection 2.4 was added to elaborate on the research hypothesis based on the literature review.

Details are as follows:

(Line 213-221) Based on the above literature review, the following hypotheses are proposed.

1) In GCH, individuals have a strong perception of the values of CES.

2) In GCH, individuals can feel stress recovery and attention restoration.

3) Certain values of CES can significantly predict recovery potential.

Furthermore, considering the findings of Riechers et al. (2018) on the differences in urban green perception among people with different CES sensitivities [54], we would also like to address the following hypothesis:

4) Perceived stress reduction and attention recovery differ for individuals with varying CES sensitivity in GCH.

Point 4:

Review your references and confirm that you have at least 60% related to articles or other types of documents that have no more than 5-6 years of publication. If there is not this proportion, please include newer references.

Response 4:

Thank you for pointing this out. After recomposing both the Introduction and Literature Review sections, we have added and removed some content and references to make the manuscript easier to read and following. Also, we have adjusted the references to ensure the number of latest references.

Point 5:

In your conclusions, green cultural heritage has the attributes of cultural heritage, education, and bequest. Aesthetic and cultural heritage significantly affect perceived stress reduction, while attention recovery shows a significant positive relationship with aesthetic value and sense of place. The results demonstrate the reliability and validity of cultural ecosystem services as a method of evaluating the environmental characteristics of green cultural heritage sites. Also, include future research and how you intend to increase the impact of your work.

Response 5:

Agree with this comment. Based on this and previous comments, we have moved the limitations and implications of the study to the Conclusion section to be consistent with this recommendation. Furthermore, for future research we would like to conduct more detailed research on the landscape elements, specific CES values and restoration experience. In this way, we hope that more stakeholders (government, scholars, citizens, etc.) would pay more attention to and protect green cultural heritage.

Details are as follows.

(Line 650-713) There are the following main research implications in this study. First, CES can be used as a useful measure of green cultural heritage. Scholars have yet to adequately mention the value of culture and history among perceptual factors in previous assessments of green space perception. Nolin (2019) argued that in the Anthropocene era, any natural place has cultural value because it has a human imprint, especially in urban environments [9385]. In terms of the cultural dimension of PSD, Peschardt and Stigsdotter (2013) described it as an environment with cultural characteristics such as fountains, sculptures, or exotic plants [6053], while Luo et al. (2022) described it as having many man-made elements [4116]. The experience of human-shaped man-made components, such as fountains, statues, and canals, are referred to as culturally relevant elements [9486] and inherent physical relics. In contrast to the simple cultural dimension, cultural heritage is a tangible and intangible asset passed down through the generations [1]. It emphasizes not only the inherent temporal dimension of all physical remains, but also the role of heritage as a source of historical knowledge, which people of different periods have used as a tool to learn more about the way of life of previous generations [9587,96]. The three values of cultural heritage, bequest, and education in CES coincide with the characteristics of cultural heritage. In the previous evaluation system, it was difficult to observe the presentation and evaluation of these values.

Second, the finding highlights the importance of preserving GCH, not only for their historical and cultural significance but also for their potential to provide mental and emotional benefits to the community. Urban planners and landscape designers may prioritize the careful conservation of such sites to maintain their unique qualities that contribute to stress reduction and attention restoration. Besides, recognizing these sites’ potential as stress-reducing and attention-restoring spaces, they could be developed as cultural and recreational hubs within cities. This could involve creating pedestrian-friendly areas, providing seating, and enhancing access to the site, encouraging people to spend more time there and engage with its cultural value.

Third, to help visitors and locals understand the green cultural heritage site's significance and its potential impact on mental well-being, interpretive signage can be in-stalled. This signage could explain the historical context, emphasize the importance of the site for mental restoration, and encourage mindful exploration. Besides, successful integration of the stress-reducing and attention-restoring elements with the site's cultural heritage requires collaboration between urban planners and heritage experts. This collaboration can lead to innovative designs that preserve the site's authenticity while enhancing its positive impact on mental well-being.

Last, by recognizing the unique qualities of a GCH as a stress-reducing and attention-restoring space, urban planning and landscape design can enhance the overall value and importance of such sites in a city. The careful consideration of historical preservation and cultural significance, alongside the well-being of residents and visitors, creates an opportunity to create culturally rich and mentally nourishing urban environments.

Several limitations apply to this study. The current scope of green cultural heritage is still vague, and our study chose only one site and just 64 respondents which may lead to some limitations, and a new round of evaluation is necessary in the future to increase validity. Second, there are some limitations of recruiting college students. However, the decision to recruit college students as participants was based on specific considerations in experimental cost. The recruitment of an adult non-student population requires significant funding in Japan; we did not have sufficient funding, and therefore recruited only a student sample in this exploratory experiment. Besides, Japan was still in the midst of a pandemic at the time of the survey, so it became very difficult to openly recruit samples, whereas it was relatively easy to organize college students from the university to conduct the survey. Therefore, data from different groups could provide various insights for this topic, and future studies that include participants of different ages, nationalities, occupations, residence times, and cultural backgrounds may be more valuable. Third, additional psychological state scales (e.g., the profile of mood states, anxiety etc.) as well as physiological indicators (e.g., blood pressure, blood oxygen levels, skin conductance levels, etc.) could be applied to measure psychological recovery from visiting green cultural heritage sites. Additionally, due to study design and conditioning limitations, we only collected data on subjective perceptions of restorative experiences, which may have some potential bias, although this is a common methodology used in previous studies. Future studies could, for example, consider measuring objective attentional improvement through pre-visit versus post-visit attention test scores. Moreover, a comparison of the restoration experience differences between green cultural heritage sites and general green spaces would be valuable. Furthermore, due to time and cost limitations, VR usage could be an effective approach to gathering data in a short time period in future studies.

 Additional clarifications

In addition to the above comments, all spelling and grammatical errors have been corrected. Besides, we checked the language of the manuscript again to ensure it was written correctly.

Once again, we thank you for the time you put in reviewing our paper and look forward to meeting your expectations. Since your inputs have been precious, in the eventuality of a publication, we would like to acknowledge your contribution explicitly.

Round 2

Reviewer 1 Report

Comments and Suggestions for Authors

Thank you for resubmitting the manuscript. I felt that the questions I raised were reasonably explained, and this manuscript was appropriately revised based on my comments. Therefore, I recommend that this manuscript could be accepted after minor revision.

Comments on the Quality of English Language

The quality of the English language in the manuscript is satisfactory.

Reviewer 2 Report

Comments and Suggestions for Authors

Dear Editor

Dear Authors

I'm pleased to see that the author has diligently addressed all the comments, including mine, and made substantial improvements to the manuscript. Given the revisions made and the enhanced quality of the work, I recommend accepting the manuscript for publication. I have no additional comments at this time.

Thank you to the authors for their responsiveness to the reviewers' suggestions.

Best regards,

Reviewer 3 Report

Comments and Suggestions for Authors

The author has made the necessary corrections and comments in response to the comments of the reviewer.